# NMDAR-dependent Argonaute 2 phosphorylation regulates miRNA activity and dendritic spine plasticity

Dipen Rajgor[1] (iD), Thomas M Sanderson[2], Mascia Amici[2], Graham L Collingridge[2,3,4] & Jonathan G Hanley[1,*] (iD)

## Abstract

MicroRNAs (miRNAs) repress translation of target mRNAs by associating with Argonaute (Ago) proteins to form the RNA-induced silencing complex (RISC), underpinning a powerful mechanism for fine-tuning protein expression. Specific miRNAs are required for NMDA receptor (NMDAR)-dependent synaptic plasticity by modulating the translation of proteins involved in dendritic spine morphogenesis or synaptic transmission. However, it is unknown how NMDAR stimulation stimulates RISC activity to rapidly repress translation of synaptic proteins. We show that NMDAR stimulation transiently increases Akt-dependent phosphorylation of Ago2 at S387, which causes an increase in binding to GW182 and a rapid increase in translational repression of *LIMK1* via miR-134. Furthermore, NMDAR-dependent down-regulation of endogenous LIMK1 translation in dendrites and dendritic spine shrinkage requires phospho-regulation of Ago2 at S387. AMPAR trafficking and hippocampal LTD do not involve S387 phosphorylation, defining this mechanism as a specific pathway for structural plasticity. This work defines a novel mechanism for the rapid transduction of NMDAR stimulation into miRNA-mediated translational repression to control dendritic spine morphology.

**Keywords** Argonaute; dendritic spine; MicroRNA; phosphorylation; synaptic plasticity; translation

**Subject Categories** Neuroscience; RNA Biology

**The EMBO Journal (2018) 37: e97943**

## Introduction

MicroRNAs (miRNAs) are small non-coding endogenous RNA molecules that repress the translation of target mRNAs through complementary binding in the transcript 3′-untranslated region (3′UTR), a process that has emerged in the past decade as being fundamentally important for fine-tuning protein synthesis in a wide range of cellular processes. Nearly half of all mammalian miRNAs are expressed at high levels in the brain, where they regulate neuronal development, spine morphogenesis, and synaptic function (Kosik, 2006; McNeill & Van Vactor, 2012; Weiss *et al*, 2015; Hu & Li, 2017). A number of miRNAs have been shown to be involved in specific forms of learning and memory, and dysfunction of miRNA systems is implicated in neurological and neuropsychiatric diseases including Alzheimer's, Huntington's, schizophrenia and drug addiction (Wang *et al*, 2012; Kocerha *et al*, 2015). Furthermore, a large proportion of neuronal miRNAs are enriched in dendrites, and several have been assigned roles in modulating the local translation of specific proteins involved in excitatory synaptic transmission or in regulating the actin cytoskeleton to control the morphology of dendritic spines, which house excitatory synapses (Lippi *et al*, 2011; Bicker *et al*, 2014; Hu *et al*, 2014, 2015; Gu *et al*, 2015).

The size and morphology of dendritic spines are dynamically regulated. Structural plasticity of spines occurs alongside plasticity of synaptic transmission, requires modifications to the underlying actin cytoskeleton and is regulated by specific kinds of synaptic activity, most notably NMDA receptor (NMDAR) stimulation (Kasai *et al*, 2010; Bosch & Hayashi, 2012; Fortin *et al*, 2012). Long-term potentiation (LTP) is an increase in synaptic strength caused by up-regulating synaptic AMPA receptors (AMPARs) and involves an increase in spine size, whereas long-term depression (LTD) is a decrease in synaptic strength caused by the internalisation of synaptic AMPARs and associated spine shrinkage (Hanley, 2008; Anggono & Huganir, 2012; Fortin *et al*, 2012). Long-term plasticity of dendritic spines is thought to be an important cellular mechanism for information storage in the brain and therefore to play an essential role in learning and memory and the fine-tuning of neural circuitry during development (Kasai *et al*, 2010; Caroni *et al*, 2012).

Long-term changes in synaptic efficacy that underlie the persistent formation of memories require changes in the synthesis of synaptic proteins by the activity-dependent local regulation of

1  Centre for Synaptic Plasticity and School of Biochemistry, University of Bristol, Bristol, UK
2  Centre for Synaptic Plasticity and School of Physiology, Pharmacology & Neuroscience, University of Bristol, Bristol, UK
3  Department of Physiology, Faculty of Medicine, University of Toronto, Toronto, ON, Canada
4  Lunenfeld-Tanenbaum Research Institute, Mount Sinai Hospital, Toronto, ON, Canada
   *Corresponding author. Tel: +44 (0)117 3311944; E-mail: jon.hanley@bristol.ac.uk

mRNA translation in dendrites close to synapses (Bramham & Wells, 2007), and a role for miRNAs in this process is emerging (Weiss *et al*, 2015). However, it is unclear how plasticity stimuli such as NMDAR stimulation are transduced into changes in miRNA activity. While the expression levels of specific miRNAs are increased in response to the induction of NMDAR-dependent chemical LTD (cLTD), it has been shown that their gene silencing activities are required for dendritic spine shrinkage or AMPAR trafficking before a detectable increase in the expression of miRNA (Hu *et al*, 2014, 2015). Therefore, the increase in expression levels of these miRNAs is not fast enough to mediate modulation of the local proteome to drive the miRNA-dependent changes in AMPAR function or spine morphology that take place soon after stimulation. Mechanisms for the rapid modulation of miRNA-dependent gene silencing in response to the induction of synaptic or structural plasticity represent a critical gap in our understanding of how protein translation is regulated in dendrites.

Argonaute (Ago) proteins are essential for miRNA-mediated gene silencing (Meister, 2013; Wilson & Doudna, 2013). MiRNAs associate with Agos in RNA-induced silencing complexes (RISCs) and guide them to target mRNAs through complementary base pairing to promote mRNA degradation or translational repression (Meister, 2013; Iwakawa & Tomari, 2015). Agos interact with numerous proteins that are essential for or modulate their gene silencing activity. In particular, GW182 (also known as TNRC6A) is an evolutionarily conserved component of RISCs and is essential for mediating the gene silencing steps downstream of RISC formation by recruiting additional proteins with relevant scaffolding or enzymatic activities (Pfaff & Meister, 2013; Jonas & Izaurralde, 2015). Importantly, Ago2 can be phosphorylated at a number of residues, some of which have been suggested to regulate RISC activity in non-neuronal cell lines by controlling Ago2-RNA or Ago2-protein interactions (Jee & Lai, 2014). Phosphorylation of Ago2 at serine 387 (S387) enhances its interaction with GW182 and increases miRNA-mediated translational repression in HeLa cells (Horman *et al*, 2013). The regulation of RISC protein–protein interactions or RISC activity by Ago2 phosphorylation remains completely unexplored in neurons, and we hypothesised that regulating critical RISC protein–protein interactions via Ago2 phosphorylation is a key mechanism to mediate the rapid modulation of miRNA-mediated gene silencing in dendrites in response to neuronal stimulation.

Here, we show that NMDAR stimulation causes an increase in phosphorylation of Ago2 at S387 by Akt, which increases Ago2 interactions with GW182 and DDX6. Phospho-regulation of Ago2 at

S387 rapidly modulates translational repression of Lim kinase 1 (LIMK1) via miR-134 and is required for NMDAR-dependent dendritic spine shrinkage, but not AMPAR trafficking.

# Results

## Interaction between Ago2 and GW182 is enhanced by NMDAR stimulation

To investigate the regulation of RISC in response to NMDAR stimulation, we focussed on the interaction between Ago2 and GW182, because this interaction is a critical regulator of RISC function (Pfaff & Meister, 2013; Jonas & Izaurralde, 2015). Co-immunoprecipitation of endogenous GW182 with endogenous Ago2 from cultured cortical neurons was significantly increased 10 min after bath application of NMDA (Fig 1A). We also analysed the association of Ago2 with the RNA helicase DDX6, which associates with Ago2 via GW182, and MOV10, which is recruited to RISC by an unknown mechanism (Meister *et al*, 2005; Chen *et al*, 2014). While Ago2-DDX6 interactions were significantly increased by NMDAR stimulation, binding to MOV10 was unaffected (Fig 1A). Ago1 interactions with GW182 and with DDx6 were unaffected by NMDAR stimulation (Fig EV1). Consistent with an increase in physical association between the two proteins, co-localisation of endogenous Ago2 and GW182 in neuronal dendrites analysed by immunocytochemistry was significantly increased by NMDAR stimulation (Fig 1B–D). Basal Ago2-GW182 co-localisation was more pronounced in the cell body compared to dendrites; however, no significant changes in the cell body were observed after NMDAR stimulation (Fig 1B and E).

## Akt-mediated phosphorylation of Ago2 at S387 is required for the NMDA-induced increase in Ago2-GW182 interaction

Previous reports suggest the Ago2-GW182 interaction is regulated in HeLa cells by phosphorylation of Ago2 at S387 by the kinase Akt (Horman *et al*, 2013; Bridge *et al*, 2017). We therefore hypothesised that the NMDAR-stimulated increase in binding would be mediated by a similar mechanism. To test this hypothesis, we investigated the effect of the specific Akt inhibitor Akti-1/2 on the Ago2-GW182 interaction. We also tested inhibitors of PKC (chelerythrine) and GSK3β (CT99021), which are kinases implicated in LTD expression (Seidenman *et al*, 2003; Peineau *et al*, 2007). The Akt inhibitor

---

**Figure 1.  Ago2 association with GW182 in neuronal dendrites increases in response to NMDAR stimulation.**

A  Endogenous Ago2-GW182 and Ago2-DDX6 interactions increase in response to NMDAR stimulation. Cortical neuronal cultures were exposed to NMDA or vehicle for 3 min, and lysates were prepared 10 min after NMDA washout and immunoprecipitated with Ago2 antibodies. Proteins were detected by Western blotting. Graph shows quantification of Ago2-GW182 interaction, normalised to vehicle control; *n* = 5. *P < 0.05; ***P < 0.001; *t*-test; mean ± SEM.

B  Analysis of endogenous Ago2-GW182 co-localisation in cortical neuronal cultures. Cortical neuronal cultures were exposed to NMDA or vehicle for 3 min, fixed 10 min after NMDA washout, permeabilised and co-stained with Ago2 and GW182 antibodies. Representative whole-cell images are shown. Scale bar = 50 μm.

C  Endogenous GW182-Ago2 co-localisation increases in response to NMDAR stimulation in neuronal dendrites. Images show dendrites taken from boxed region in (B), above. Graph shows Pearson's co-localisation coefficients; *n* = 4 independent experiments (18–24 cells per condition). *P < 0.05, *t*-test. Scale bar = 10 μm. Mean ± SEM.

D  Line-scan analyses of Ago2 and GW182 fluorescence intensities in control and NMDA-stimulated dendrites shown in (C).

E  NMDAR stimulation has no effect on endogenous Ago2-GW182 co-localisation in neuronal cell bodies. Images show cell bodies taken from boxed region in (B). Graph shows Pearson's co-localisation coefficients; *n* = 4 independent experiments (18–20 cells per condition), *t*-test. Scale bar = 10 μm. Mean ± SEM.

Source data are available online for this figure.

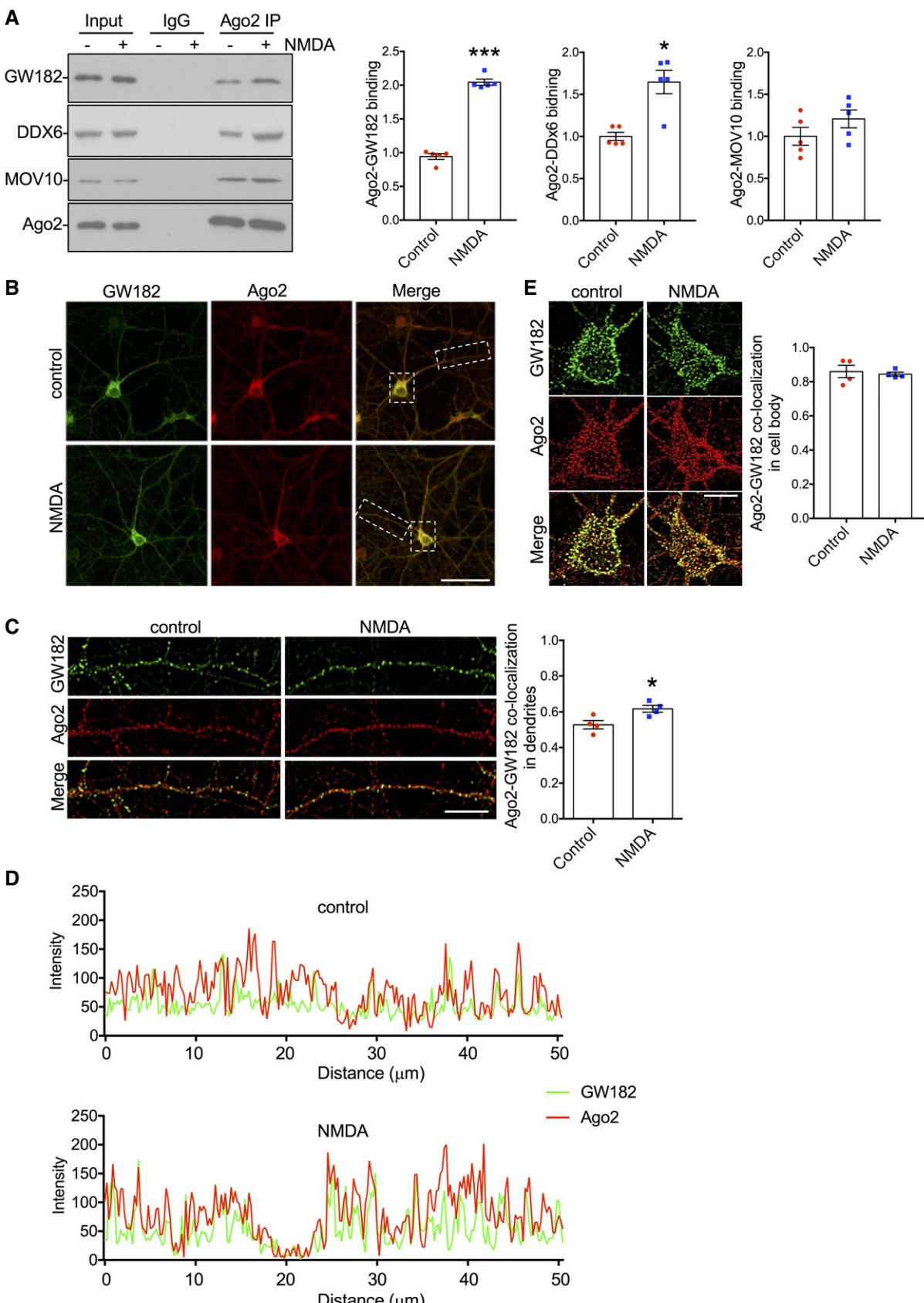

**Figure 1.**

Akti-1/2 completely blocked the NMDA-induced increase in Ago2-GW182 binding, while chelerythrine and CT99021 had no effect (Fig 2A). Next, we analysed Ago2 phosphorylation at S387 using a phospho-specific antibody. NMDAR activation caused a significant increase in S387 phosphorylation, which was blocked by Akti-1/2, but not by chelerythrine or CT99021 (Fig 2B). Interestingly, Akt inhibition reduced Ago2 phosphorylation and Ago2-GW182 interaction under unstimulated conditions, suggesting that Akt is basally active to phosphorylate S387 and promote GW182 binding to Ago2 (Fig 2A and B). These results strongly suggest that Ago2 phosphorylation and the increase in GW182-Ago2 interaction are caused by NMDAR-dependent Akt activation. To provide further support for this mechanism, we tested the effect of a second Akt inhibitor, KP372-1 and also an Akt activator, sc79. KP372-1 had a similar effect as Akti-1/2, blocking both the NMDAR-stimulated increase in Ago2 phosphorylation at S387, and the increase in Ago2-GW182 binding (Fig 2C and D). In contrast, sc79 caused an increase in S387 phosphorylation and Ago2-GW182 interaction under basal conditions, which occluded the effect of NMDA (Fig 2C and D). The p38 MAPK pathway has also been shown to phosphorylate Ago2 at S387 in non-neuronal cell lines (Zeng *et al*, 2008), so we analysed Ago2-GW182 binding and S387 phosphorylation in the presence of the p38 MAPK inhibitor SB203580. In contrast to Akti-1/2, SB203580 did not affect the NMDAR-dependent increase in GW182 binding or S387 phosphorylation (Fig 2E and F). Taken together, these results demonstrate that phosphorylation of Ago2 at S387 and Ago2 binding to GW182 are increased by NMDAR stimulation in an Akt-dependent manner.

To test directly whether the NMDAR-dependent increase in Ago2-GW182 binding is caused by Ago2 phosphorylation at S387, we generated molecular replacement constructs that express Ago2 shRNA as well as GFP or GFP-tagged shRNA-resistant Ago2. In addition to wild-type (WT) Ago2, we made constructs to express a phospho-null (S387A) or a phospho-mimic (S387D) mutant, hypothesising that the S387A mutant would behave in a similar manner as dephosphorylated Ago2, while S387D would show similar properties as phosphorylated

Ago2. Appendix Fig S1 shows that the Ago2 shRNA efficiently knocked down endogenous Ago2 to ∼ 23% of control levels. Co-expression of shRNA-resistant GFP-WT, GFP-S387A or GFP-S387D resulted in a slight over-rescue of Ago2 expression, which was ∼ 30% higher than endogenous Ago2 under control conditions, and all three recombinant proteins were expressed at the same level (Appendix Fig S1). To investigate the effect of S387 phosphorylation mutants on GW182 binding, we performed GFP-trap pull-downs on lysates prepared from cortical neurons transfected with control or molecular replacement constructs, and analysed endogenous GW182 binding to GFP-WT-Ago2, GFP-S387A-Ago2 and GFP-S387D-Ago2. Under unstimulated conditions, GFP-S387D-Ago2 showed a dramatic increase in binding to endogenous GW182 compared to GFP-WT-Ago2, while GFP-S387A-Ago2 bound GW182 slightly less than WT, which did not reach statistical significance (Fig 3A). While NMDAR stimulation caused a similar increase in GW182 binding to GFP-WT-Ago2 compared to endogenous Ago2, NMDA had no effect on GW182 binding to either GFP-S387A-Ago2 or GFP-S387D-Ago2. Hence, the S387A mutation blocks the effect of NMDAR stimulation, whereas S387D mimics and occludes the effect of NMDA. All three mammalian GW182 isoforms (TNRC6A-C) showed similar patterns of binding to Ago2 S387 mutants in heterologous cells, suggesting that the binding of Ago2 to all three isoforms is subject to regulation by phosphorylation at S387 (Fig EV2). We also analysed the interactions between GFP-Ago2 and endogenous MOV10 and DDX6 in neurons. While the interaction between Ago2 and MOV10 was unaffected by S387 mutations, association with DDX6 showed a similar pattern as GW182 binding (Fig 3A). We previously described an interaction between Ago2 and the BAR domain protein PICK1, which dissociates in response to NMDAR stimulation (Antoniou *et al*, 2014; Rajgor *et al*, 2017). This interaction was unaffected by S387 mutations in Ago2 (Fig 3A). Furthermore, NMDAR-dependent Ago2 phosphorylation at S387 was unaffected by molecular replacement of PICK1 with GFP-PICK1-AX10, a mutant whose interaction with Ago2 is resistant to NMDAR stimulation (Rajgor *et al*, 2017;

---

**Figure 2.   NMDAR-dependent increase in Ago2-GW182 interaction and Ago2 phosphorylation at S387 requires Akt activity.**

A   NMDAR-stimulated increase in Ago2-GW182 interaction is Akt-dependent. Cortical neuronal cultures were treated with Akti-1/2 (Akt inhibitor), chelerythrine (Chel, PKC inhibitor) or CT99021 (GSK-3β inhibitor) 20 min before NMDA or vehicle application. Lysates were prepared 10 min after NMDA washout and immunoprecipitated with Ago2 antibodies or control IgG as shown. Proteins were detected by Western blotting. The inputs are shown in (B). Graph shows quantification of Ago2-GW182 interaction, normalised to vehicle control; *n* = 3.

B   NMDAR-stimulated increase in Ago2 phosphorylation at S387 is Akt-dependent. The same lysates from (A) (1% of input) were analysed by Western blotting using antibodies against pS387 Ago2, Ago2, GW182 and GAPDH as a loading control. Graph shows quantification of pS387 Ago2 levels normalised to total Ago2; *n* = 3.

C   Further evidence that the NMDAR-stimulated increase in Ago2-GW182 interaction is Akt-dependent. Cortical neuronal cultures were treated with Akti-1/2, KP372-1 (Akt inhibitor) or sc79 (Akt activator) 20 min before NMDA or vehicle application. Lysates were prepared 10 min after NMDA washout and immunoprecipitated with Ago2 antibodies or control IgG as shown. Proteins were detected by Western blotting. The inputs are shown in (D). Graph shows quantification of Ago2-GW182 interaction, normalised to vehicle control; *n* = 4.

D   Further evidence that the NMDAR-stimulated increase in Ago2 phosphorylation at S387 is Akt-dependent. The same lysates from (C) (1% of input) were analysed by Western blotting. Graph shows quantification of pS387 Ago2 levels normalised to total Ago2; *n* = 4.

E   NMDAR-stimulated increase in Ago2-GW182 interaction is p38 MAP kinase-independent. Cortical neuronal cultures were treated with Akti-1/2 or SB203580 (p38MAPK inhibitor) 20 min before NMDA or vehicle application. Lysates were prepared 10 min after NMDA washout and immunoprecipitated with Ago2 antibodies or control IgG as shown. Proteins were detected by Western blotting. The inputs are shown in (F). Graph shows quantification of Ago2-GW182 interaction, normalised to vehicle control; *n* = 4.

F   NMDAR-stimulated increase in Ago2 phosphorylation at S387 is p38 MAP kinase-independent. The same lysates from (E) (1% of input) were analysed by Western blotting. Graph shows quantification of pS387 Ago2 levels normalised to total Ago2; *n* = 4.

Data information: *$P < 0.05$; **$P < 0.01$; ***$P < 0.001$; two-way ANOVA, Bonferroni *post hoc* test. Mean ± SEM.
Source data are available online for this figure.

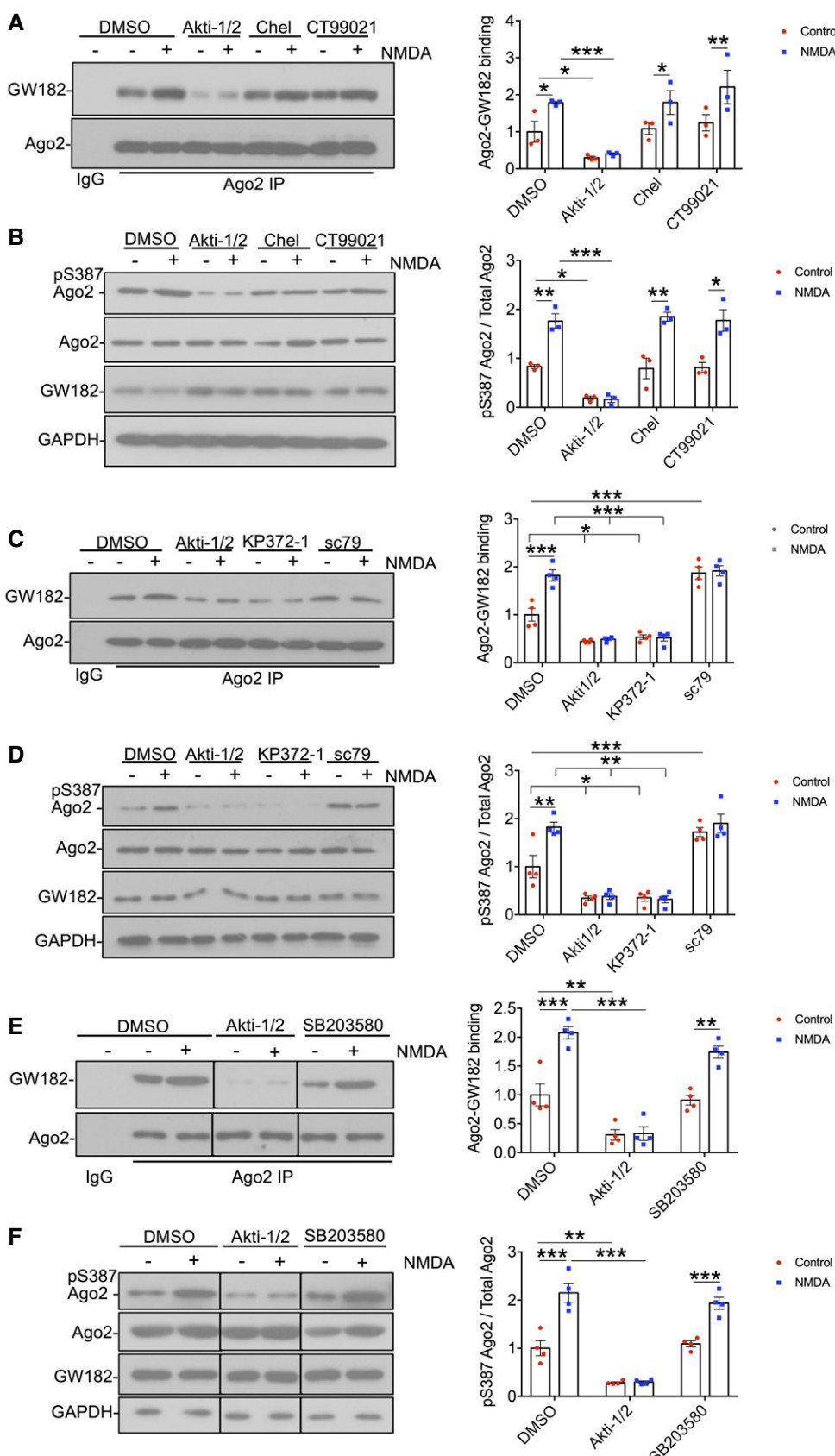

Figure 2.

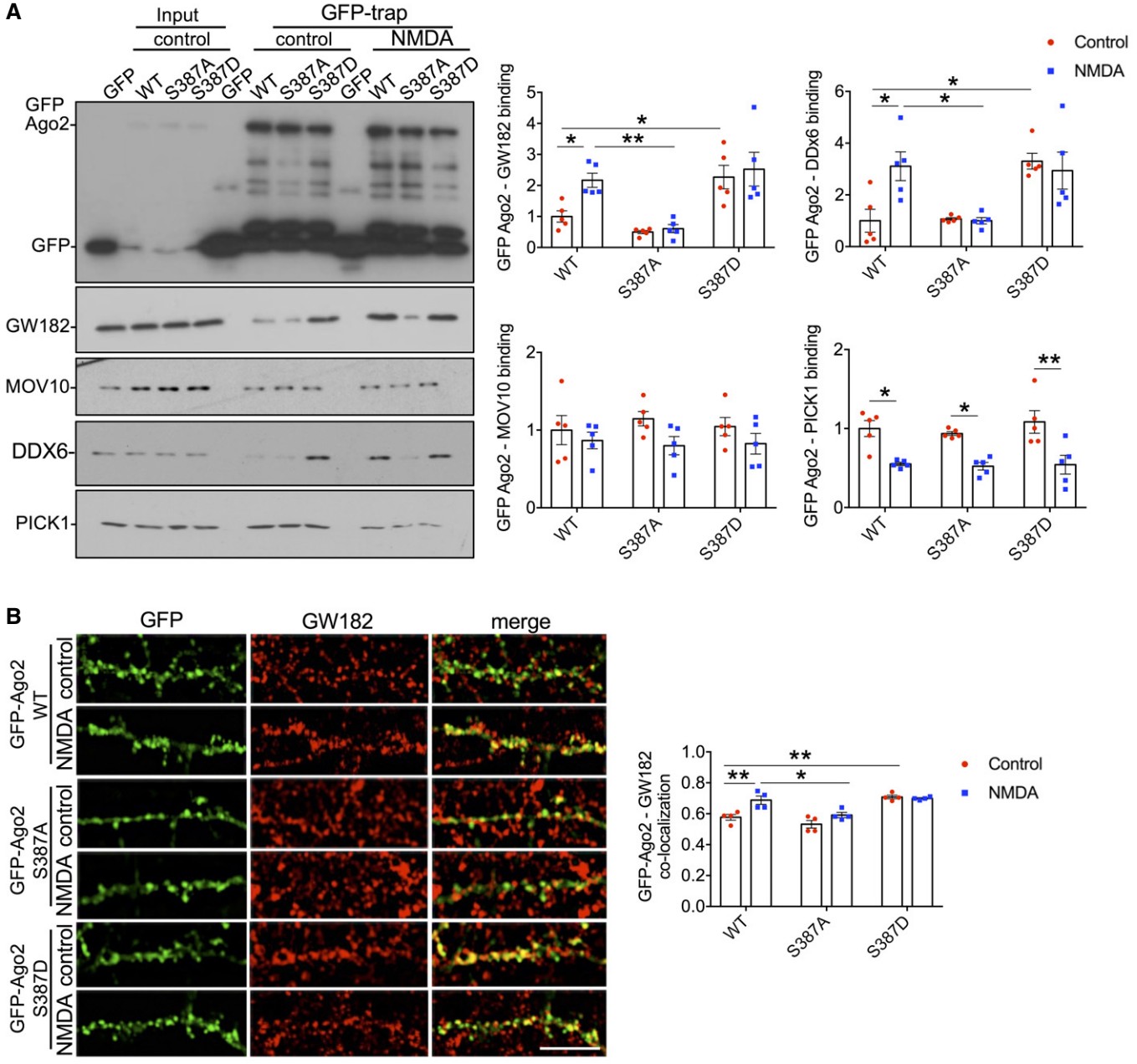

**Figure 3.  NMDA-induced increase in interaction with GW182 is caused by Ago2 phosphorylation at S387.**

A   Ago2 S387A mutation blocks, and S387D mutation occludes NMDA-induced increases in GW182 and DDX6 interactions. Cortical neurons were transfected with molecular replacement constructs expressing Ago2 shRNA plus shRNA-resistant GFP-Ago2 (WT, S387A or S387D). Lysates were prepared 10 min after NMDA washout, and GFP-Ago2 complexes were precipitated using GFP-trap beads. Bound proteins were detected by Western blotting using GFP, GW182, MOV10, DDX6 or PICK1 antibodies as shown. Graphs show quantification of GFP-Ago2 interactions, normalised to untreated WT condition; $n = 5$. *$P < 0.05$, **$P < 0.01$; two-way ANOVA, Bonferroni *post hoc* test. Mean ± SEM.

B   Ago2 S387A mutation blocks, and S387D mutation occludes NMDA-induced increase in GW182 co-localisation in neuronal dendrites. Cortical neurons were transfected with molecular replacement constructs expressing Ago2 shRNA plus shRNA-resistant GFP-Ago2 (WT, S387A or S387D), fixed 10 min after NMDA washout, permeabilised and stained with GW182 and GFP antibodies. Graph shows Pearson's co-localisation coefficients; $n = 4$ independent experiments (11 cells per condition). *$P < 0.05$, **$P < 0.01$; two-way ANOVA, Bonferroni *post hoc* test. Scale bar = 10 μm. Mean ± SEM.

Source data are available online for this figure.

Appendix Fig S2). This suggests that NMDAR-dependent phosphorylation of Ago2 at S387 does not depend on prior dissociation from PICK1.

In a complementary set of experiments, we used confocal imaging of the same GFP-tagged Ago2 constructs to analyse their co-localisation with endogenous GW182. While all three

GFP-Ago2 proteins co-localised with GW182 puncta, GFP-S387D-Ago2 showed significantly more co-localisation than GFP-WT-Ago2, and only GFP-WT-Ago2 showed an increase in co-localisation following NMDAR stimulation (Fig 3B).

These results demonstrate that Ago2-GW182 interactions in neuronal dendrites are dynamically regulated by phosphorylation of Ago2 at S387 in response to NMDAR stimulation.

### Transient increase in GW182-Ago2 interaction and Ago2 phosphorylation at S387 in response to NMDAR stimulation

Our results so far indicate that the increases in Ago2 phosphorylation and GW182 binding take place within 10 min after NMDAR stimulation. To better understand the time course of these changes following stimulation, we analysed Ago2 phosphorylation at S387 and endogenous Ago2-GW182 binding at 0, 3, 6, 10 and 20 min after stimulation. Ago2-GW182 binding transiently increased following NMDAR stimulation, with a peak at 6 min after stimulation (Fig 4A). The increase in Ago2 phosphorylation at S387 showed a similar time course with maximum phosphorylation at 6 min (Fig 4B). Both of these changes remained significantly elevated at 10 min after stimulation and returned to baseline levels by 20 min. These results demonstrate that GW182-Ago2 binding is transiently enhanced by NMDAR stimulation, and the strengthened complex lasts for approximately 10 min. Our results presented in Fig 2 suggest that Akt is activated in response to NMDAR stimulation. We tested this directly using an Akt phospho-specific antibody against pS473, which is a well-established marker for activated Akt (Perkinton *et al*, 2002; Sutton & Chandler, 2002). NMDAR stimulation caused a similar transient increase in Akt activation, which peaked slightly earlier, at 3 min after stimulation (Fig 4B), consistent with a mechanism in which Akt activity is upstream of Ago2 phosphorylation and GW182 binding.

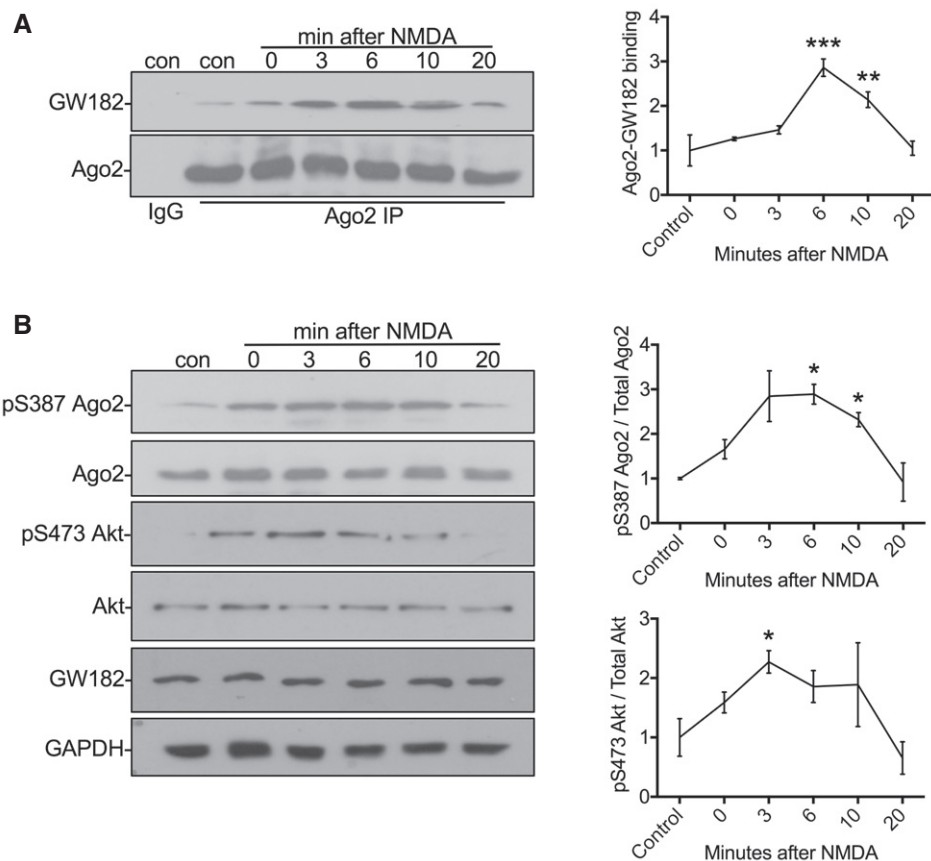

**Figure 4.  Transient increase in GW182-Ago2 interaction and S387 phosphorylation in response to NMDAR stimulation.**

A   Transient increase in Ago2-GW182 interaction. Cortical neuronal cultures were exposed to NMDA or vehicle for 3 min, and lysates were prepared 0, 3, 6, 10, 20 min after NMDA washout and immunoprecipitated with Ago2 antibodies or control IgG. Proteins were detected by Western blotting. The inputs are shown in (B). Graph shows quantification of Ago2-GW182 interaction, normalised to vehicle control; *n* = 4. **P < 0.01, ***P < 0.001; one-way ANOVA, Bonferroni *post hoc* test. Mean ± SEM.

B   Transient increase in S387 phosphorylation and Akt activation. The same lysates from (A) (1% of input) were analysed by Western blotting using antibodies against pS387 Ago2, Ago2, pS473 Akt, Akt, GW182 and GAPDH as a loading control. Graphs show quantification of pS387 Ago2 levels normalised to total Ago2 (top) and pS473 Akt normalised to total Akt (bottom); *n* = 4. *P < 0.05; two-way ANOVA, Bonferroni *post hoc* test. Mean ± SEM.

Source data are available online for this figure.

## NMDAR-dependent translational repression via miR-134 is regulated by Ago2 phosphorylation at S387

To investigate the effect of increasing the pS387-dependent Ago2-GW182 interaction on miRNA-mediated translational repression, we employed dual-luciferase assays, with *Renilla* control and *Firefly* reporter constructs incorporating 3′UTRs of known targets of endogenous miRNAs. In these assays, a decrease in luciferase activity represents an increase in miRNA-mediated translational repression (and vice versa). We analysed two dendritically regulated UTRs; *LIMK1*, which is regulated by miR-134 (Schratt *et al*, 2006), and *APT1*, which is regulated by miR-138 (Siegel *et al*, 2009). Both of these miRNAs have been shown previously to regulate dendritic spine morphology (Schratt *et al*, 2006; Siegel *et al*, 2009), and we previously demonstrated that NMDAR activation increased translational repression of the *LIMK1* reporter via miR-134 and of the *APT1* reporter via miR-138 within 10 min after stimulation (Antoniou *et al*, 2014; Rajgor *et al*, 2017). Knockdown of Ago2 by shRNA caused a dramatic increase in expression of both reporter constructs, consistent with a deficit of miRNA-mediated translational repression, and NMDAR stimulation had no effect under these conditions (Fig 5A and B). The expression levels of Ago1 and Ago3 were unaffected by Ago2 knockdown (Appendix Fig S3A), indicating that these other isoforms were not up-regulated to compensate for the loss of Ago2. Co-expression of sh-resistant GFP-WT-Ago2 fully rescued both the basal level of luciferase reporter expression, as well as the sensitivity to NMDAR stimulation (Fig 5A and B), indicating that the slight overexpression of Ago2 with the molecular replacement constructs (see Appendix Fig S1) had no functional impact on miRNA activity. Co-expression of Ago1 did not rescue the functional deficit caused by Ago2 knockdown (Appendix Fig S3B). Interestingly, molecular replacement with S387 mutants had distinct effects on *LIMK1* silencing by miR-134 compared to *APT1* silencing via miR-138. GFP-S387A-Ago2 expression caused a significant increase in basal expression of the *LIMK1* reporter, suggesting reduced RISC activity, whereas GFP-S387D-Ago2 expression caused a significant decrease in *LIMK1* reporter expression, indicative of increased RISC activity. Both S387 mutants abolished NMDAR-dependent changes in *LIMK1* reporter expression (Fig 5A). These results indicate that the NMDA-induced regulation of *LIMK1* silencing via miR-134 depends on the dynamic phosphorylation of Ago2 at S387. In contrast, basal expression of the *APT1* reporter and sensitivity to NMDA were unaffected by the S387 mutations; cultures expressing molecular replacement constructs for GFP-S387A-Ago2, GFP-S387D-Ago2 and GFP-WT-Ago2 all showed patterns of luciferase expression under basal and stimulated conditions that were indistinguishable from controls (Fig 5B). This indicates that translational repression of *APT1* via miR-138 activity is regulated by a mechanism that does not require S387 phosphorylation. In addition, we analysed a further luciferase reporter construct incorporating the *LIN41* 3′UTR, which is a target for the miRNA Let7. In contrast to miR-134 and miR-138, Let7 has not been shown to be targeted to dendrites. While expression of the *LIN41* reporter was increased by Ago2 shRNA, it was unaffected by NMDAR activation and unaffected by S387 mutation (Fig 5C).

These results suggested that the regulation of translational repression via S387 phosphorylation depends either on a specific property of miR-134, or alternatively on a specific property of the *LIMK1* 3′UTR. To investigate this further, we analysed the expression of luciferase reporters incorporating the 3′UTRs of *PUM2* and *CREB1*, which have been shown previously to be regulated by miR-134 (Fiore *et al*, 2014). Interestingly, while both reporters were sensitive to Ago2 shRNA, neither was affected by molecular replacement with S387 mutants (Fig 5D and E). This suggests that specific characteristics of the *LIMK1* 3′UTR, which are distinct from the *CREB1* and *PUM2* 3′UTRs, are necessary for translational regulation via Ago2 phosphorylation at S387.

Importantly, to confirm that the observed changes in reporter expression were miRNA-dependent, we performed parallel experiments with reporter constructs carrying mutations in the miR-134, miR-138 or Let7 seed regions of *LIMK1*, *APT1* or *LIN41* 3′UTRs, respectively, to prevent miRNA binding (Storchel *et al*, 2015; Fig EV3A–C; Appendix Fig S4A–C). Indeed, all translational control over the luciferase constructs was abolished by these mutations, supporting our conclusions that the observed changes are caused by altered miR-134, miR-138 and Let-7 activity, respectively. Interestingly, while reporter constructs incorporating *PUM2* and *CREB1* 3′UTRs with mutated miR-134 seed regions were insensitive to NMDAR stimulation and to S387 mutations, Ago2 shRNA did cause an increase in luciferase activity (Fig EV3F and G; Appendix Fig S4D and E). This suggests that *PUM2* and *CREB1* are subject to regulation by additional miRNAs apart from miR-134, whereas *LIMK1* translation is selectively repressed by miR-134.

Since our results presented in Fig 2 suggest that Akt phosphorylates S387 in response to NMDAR stimulation, we analysed the effect of Akt inhibition on luciferase reporter expression. Both Akt inhibitors Akti-1/2 and KP372-1 caused a decrease in basal silencing of *LIMK1* via miR-134 and also blocked the NMDAR-stimulated increase in miRNA activity, while PKC inhibition (chelerythrine) and GSK3β inhibition (CT99021) had no effect (Fig 5F). In contrast, miR-138 activity was unaffected by any of these inhibitors (Fig 5G), which is consistent with our observations that S387 mutants had no effect on *APT1* silencing via miR-138. None of the inhibitors affected expression of luciferase reporter constructs carrying mutations in the 3′UTR to block binding of miRNA (Fig EV3D and E).

Taken together, these results strongly suggest that NMDAR stimulation causes a rapid up-regulation of miR-134-dependent *LIMK1* silencing via Akt-mediated phosphorylation of Ago2 at S387. *APT1* silencing via miR-138 activity is also up-regulated by NMDAR stimulation, but is controlled by an alternative mechanism.

## Phospho-regulation of Ago2 at S387 mediates NMDA-stimulated translational repression of endogenous LIMK1 in dendrites

LIMK1 is a key component of a signalling pathway that controls actin polymerisation in several cellular processes, including dendritic spine morphogenesis (Hotulainen & Hoogenraad, 2010; Fortin *et al*, 2012). It has been shown previously that LIMK1 deletion results in smaller spine heads, and LIMK1 overexpression blocks NMDA-induced shrinkage of dendritic spines (Meng *et al*, 2002; Calabrese *et al*, 2014), suggesting that LIMK1 plays an important role in maintaining spine size, or restricting spine shrinkage. A previous report demonstrated that endogenous LIMK1 protein expression is down-regulated by miR-134 in neurons (Schratt *et al*, 2006), and our luciferase reporter assay results suggest that *LIMK1* translation is regulated by an NMDAR-dependent mechanism that

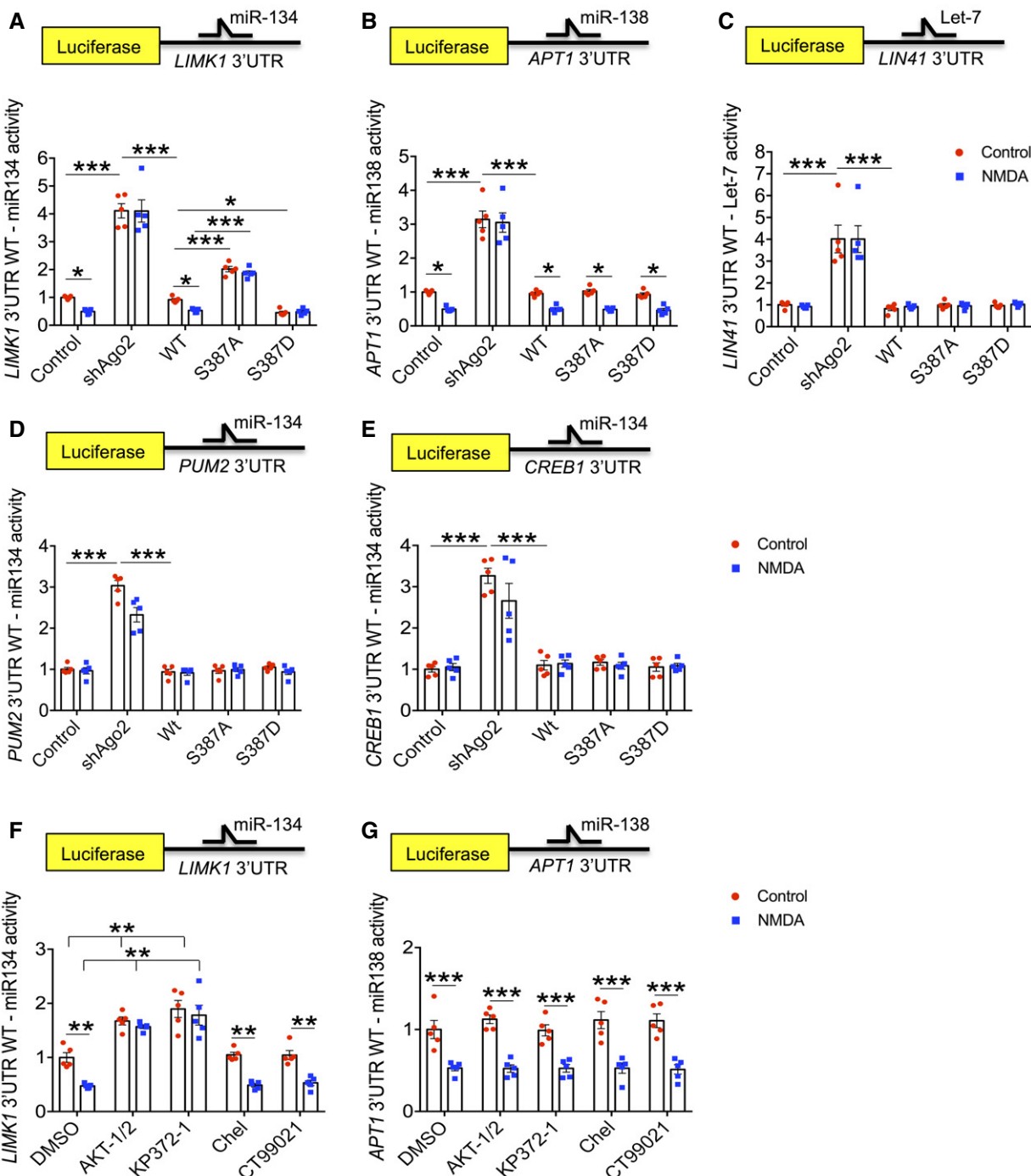

**Figure 5.  MiR-134-mediated *LIMK1* silencing is regulated by Akt-dependent Ago2 phosphorylation at S387 in response to NMDAR stimulation.**

A–C    Silencing of *LIMK1* via miR-134 (A), but not *APT1* via miR-138 (B) or *LIN41* via Let7 (C), is regulated by S387 phosphorylation. Cultured cortical neurons transfected with Ago2 molecular replacement constructs as well as *Renilla* luciferase and *Firefly* luciferase reporters containing *LIMK1* (A), *APT1* (B) or *LIN41* (C) 3′UTRs were treated with NMDA or vehicle for 3 min. 10 min after NMDA washout, lysates were prepared for dual-luciferase assays.

D, E    Silencing of miR-134 targets *PUM2* and *CREB1* is unaffected by S387 phosphorylation and NMDAR stimulation. Cultured cortical neurons transfected with Ago2 molecular replacement constructs as well as *Renilla* luciferase and *Firefly* luciferase reporters containing *PUM2* (D), *CREB1* (E) 3′UTRs were treated with NMDA or vehicle for 3 min. 10 min after NMDA washout, lysates were prepared for dual-luciferase assays.

F, G    NMDAR-dependent silencing of *LIMK1* via miR-134 (D), but not *APT1* via miR-138 (E), is Akt-dependent. Cultured cortical neurons transfected with Ago2 molecular replacement constructs as well as *Renilla* luciferase and *Firefly* luciferase reporter containing *LIMK1* (D) or *APT1* (E) 3′UTR were treated with inhibitors as shown 20 min before NMDA or vehicle. 10 min after NMDA washout, lysates were prepared for dual-luciferase assays.

Data information: All graphs show *Luciferase/Renilla* ratios, normalised to vehicle control; $n = 5$. *$P < 0.05$; **$P < 0.01$; ***$P < 0.001$; two-way ANOVA, Bonferroni *post hoc* test. Mean ± SEM.

involves Ago2 phosphorylation at S387. We therefore hypothesised that increased RISC activity caused by Ago2 phosphorylation at S387 is involved in NMDA-stimulated spine shrinkage via translational repression of LIMK1.

To test this hypothesis, we asked whether endogenous LIMK1 is regulated in a similar manner as the *LIMK1* 3′UTR luciferase reporter. We initially performed a time course experiment to investigate LIMK1 expression after NMDAR stimulation. Consistent with our luciferase assays, we observed a significant reduction in LIMK1 protein within the first 10 min after stimulation, which was further diminished to < 50% of control levels by 40 min (Fig 6A). Furthermore, inhibition of Akt with Akti-1/2 caused a significant block of NMDAR-dependent down-regulation of LIMK1 expression at 40 min (Fig 6B), demonstrating a requirement for Akt activation, and suggesting a role for regulation of Ago2 by S387 phosphorylation. In the light of our striking luciferase assay results, we also analysed expression of endogenous APT1 (acyl protein thioesterase 1), which is involved in dendritic spine morphogenesis by modulating the palmitoylation of synaptic proteins (Siegel *et al*, 2009). In agreement with our luciferase assays, APT1 protein expression was reduced by NMDAR stimulation (Fig EV4A), while Akt inhibition had no effect (Fig EV4B).

To directly investigate a role for Ago2 phosphorylation at S387, we used immunocytochemistry to analyse LIMK1 protein in dendrites of neurons transfected with Ago2 molecular replacement constructs. In control neurons expressing GFP, NMDAR stimulation caused a significant decrease in LIMK1 expression 40 min after stimulation (Fig 6C). In neurons expressing Ago2 shRNA, LIMK1 expression under basal conditions was increased compared to controls and was unaffected by NMDAR stimulation (Fig 6C). ShRNA-resistant GFP-WT-Ago2 rescued basal LIMK1 levels and the NMDAR-stimulated decrease in expression seen in control neurons. Molecular replacement with GFP-S387A-Ago2 caused a small but significant increase in basal LIMK1 expression, while GFP-S387D-Ago2 caused a corresponding decrease in LIMK1. NMDA had no effect on LIMK1 expression in neurons expressing either S387 mutant (Fig 6C). We carried out equivalent experiments to analyse NMDAR-dependent changes in APT1 protein expression in neuronal dendrites expressing Ago2 S387 mutants. While dendritic APT1 was reduced by NMDAR stimulation, S387 mutations had no effect (Fig EV4C). These results strongly suggest that an NMDAR-stimulated increase

in translational repression via Ago2 S387 phosphorylation causes a reduction in endogenous LIMK1 expression, but not APT1 expression, in neuronal dendrites.

To investigate *de novo* synthesis of endogenous LIMK1, we used puromycin-proximity ligation assays (Puro-PLA), which exploits the incorporation of puromycin into newly synthesised polypeptides, which can be labelled with an anti-puromycin antibody at the same time as an antibody against LIMK1. A PLA signal reports the close (< 5 nm) proximity of puromycin and LIMK1 antibodies and hence a site of newly synthesised protein (tom Dieck *et al*, 2015; Sambandan *et al*, 2017). Puro-LIMK1 PLA puncta were readily detectable in the dendrites of cultured neurons incubated with puromycin. In contrast, PLA signal was absent in control cells that were not exposed to puromycin (Fig 6D). NMDA caused a significant reduction in the number of puncta at 40 min after stimulation, demonstrating that endogenous LIMK1 translation is reduced by NMDAR stimulation within this time frame. Ago2 knockdown by shRNA caused a significant increase in the density of Puro-LIMK1 PLA puncta in dendrites, consistent with a reduction in miRNA-mediated translational repression. NMDA had no effect on the density of puncta in neurons expressing Ago2 shRNA, indicating that the NMDAR-dependent reduction in LIMK1 translation is miRNA-dependent. Co-expression of sh-resistant GFP-WT-Ago2 fully rescued both the basal density of Puro-LIMK1 PLA puncta, as well as the sensitivity to NMDAR stimulation. Molecular replacement with GFP-S387A-Ago2 caused no change in basal puncta density, but completely blocked the effect of NMDA, indicating that Ago2 phosphorylation at S387 is required for the NMDAR-dependent repression of endogenous LIMK1 translation. This is further supported by GFP-S387D-Ago2 expression, which caused a reduction in Puro-LIMK1 PLA puncta under basal conditions and occluded the effect of subsequent NMDAR stimulation (Fig 6D). These experiments demonstrate that the synthesis of LIMK1 in neuronal dendrites is down-regulated in a miRNA-dependent manner via a mechanism that requires Ago2 phosphorylation at S387.

### Phospho-regulation of Ago2 at S387 is required for NMDA-induced spine shrinkage

Next, we analysed NMDA-stimulated dendritic spine shrinkage on cultured neurons transfected with molecular replacement constructs

---

**Figure 6.  NMDAR stimulation regulates dendritic LIMK1 translation via Akt activation and Ago2 phosphorylation at S387.**

A  Endogenous LIMK1 protein levels are rapidly reduced in response to NMDAR stimulation. Cortical neuronal cultures were exposed to NMDA or vehicle for 3 min, and lysates were prepared 10, 20 or 40 min after NMDA washout and analysed by Western blotting. Graphs show quantification of LIMK1 expression normalised to vehicle control; *n* = 6. *$P < 0.05$; **$P < 0.01$; one-way ANOVA, Bonferroni *post hoc* test. Mean ± SEM.

B  NMDAR-dependent decrease in LIMK1 is Akt-dependent. Cortical neuronal cultures were treated with Akti-1/2 20 min before NMDA or vehicle application, and lysates were prepared 40 min after NMDA washout and analysed by Western blotting. Graphs show quantification of LIMK1 expression normalised to vehicle control; *n* = 5. **$P < 0.01$; ***$P < 0.001$ one-way ANOVA, Bonferroni *post hoc* test. Mean ± SEM.

C  NMDAR-dependent decrease in dendritic LIMK1 expression requires Ago2 phosphorylation at S387. Cortical neurons were transfected with molecular replacement constructs expressing Ago2 shRNA plus shRNA-resistant GFP-Ago2 (WT, S387A or S387D), fixed 40 min after NMDA washout, permeabilised and stained with LIMK1 antibodies (red channel). GFP signal was maximised at acquisition so that dendrites could be effectively visualised. Graph shows LIMK1 staining intensity in dendrites normalised to vehicle control; *n* = 5 independent experiments (10 cells per condition). *$P < 0.05$; **$P < 0.01$, two-way ANOVA, Bonferroni *post hoc* test. Scale bar = 10 μm. Mean ± SEM.

D  *De novo* translation of endogenous LIMK1 is regulated in dendrites by NMDAR stimulation via Ago2 phosphorylation at S387. Cortical neuronal cultures were transfected with molecular replacement constructs expressing Ago2 shRNA plus shRNA-resistant GFP-Ago2 (WT, S387A or S387D). Neurons were incubated with 1 μM puromycin in the presence or absence of 50 μM NMDA for 3 min. Following NMDA washout, neurons were incubated with puromycin for 40 min, after which the cells were fixed and processed for Puro-LIMK1 PLA (see Materials and Methods). Puro-PLA was also performed on GFP-transfected neurons that were not incubated with puromycin as a negative control (bottom row of images). *n* = 5 independent experiments (10–13 cells per condition). **$P < 0.01$; ***$P < 0.001$ two-way ANOVA, Bonferroni *post hoc* test. Scale bar = 10 μm. Mean ± SEM.

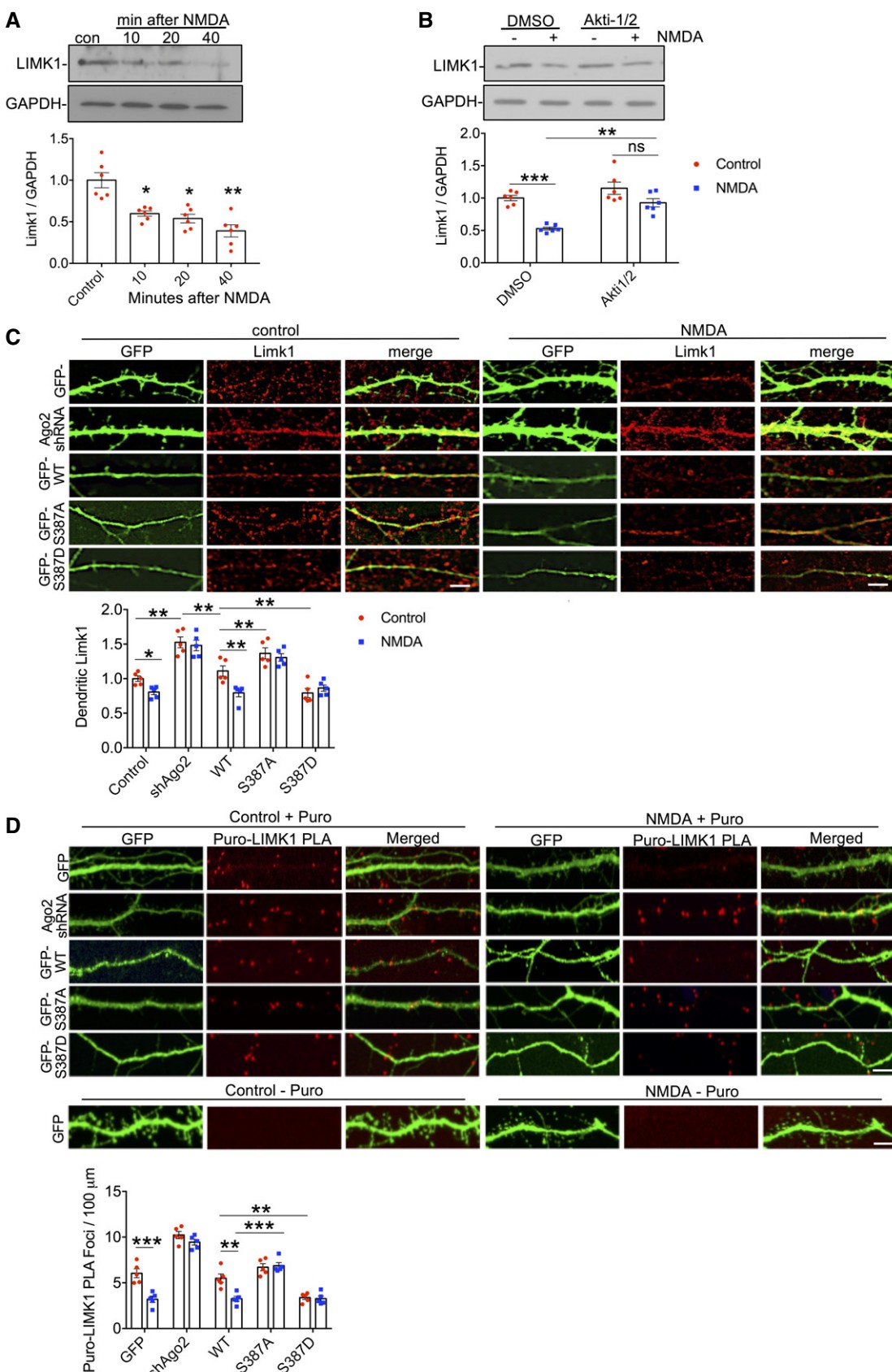

**Figure 6.**

to express GFP-tagged Ago2 S387 mutants. In control cells expressing GFP, NMDAR activation caused spine shrinkage of ~ 25% at 40 min after stimulation (Fig 7A). Surprisingly, Ago2 knockdown by shRNA had no effect on basal spine size, nor on NMDA-stimulated spine shrinkage (Fig 7A). However, molecular replacement with GFP-S387A-Ago2 caused a significant increase in basal spine size and GFP-S387D-Ago2 caused a significant decrease in basal spine size. Furthermore, NMDA-stimulated spine shrinkage was abolished in neurons expressing either mutant (Fig 7A). These results show that phosphorylation of Ago2 at S387 is an essential component of NMDAR-dependent dendritic spine shrinkage, suggesting that Akt activity is required for this process. To test this directly, we applied the Akt inhibitor Akti-1/2 during and after NMDAR stimulation and analysed spine size 40 min later. Akti-1/2 completely blocked NMDA-induced spine shrinkage (Fig 7B).

Analysis of spine density indicated that NMDAR stimulation had no significant effect under these experimental conditions. However, Ago2 shRNA did cause a significant increase in spine density, a phenotype that was fully rescued by GFP-WT-Ago2, GFP-S387A-Ago2 or GFP-S387D-Ago2 (Fig 7A), suggesting that S387 phosphorylation is not involved in regulating spine density.

Taken together, these results demonstrate that phospho-regulation of Ago2 at S387 by Akt is required for NMDA-induced spine shrinkage, but is not involved in regulating spine density. Considering these data in conjunction with our results for LIMK1 translation (Figs 5 and 6) leads to the hypothesis that the down-regulation of LIMK1 synthesis is a critical component of the mechanism that underlies NMDA-induced spine shrinkage. To test this, we used a myc-tagged LIMK1 construct that does not contain the native *LIMK1* 3′UTR and is therefore resistant to regulation by miR-134. NMDA-induced spine shrinkage is abolished in neurons expressing myc LIMK1 (Fig 7C), strongly suggesting that a reduction in LIMK1 expression is required for spine shrinkage.

### Phospho-regulation of Ago2 at S387 is not involved in NMDAR-stimulated AMPAR trafficking

In addition to spine shrinkage, LTD involves a removal of AMPARs from synapses, caused by increased receptor endocytosis from the cell surface and regulation in the endosomal system (Anggono & Huganir, 2012). Since our results demonstrate that NMDAR-dependent

phosphorylation of Ago2 is required for spine shrinkage, we also investigated whether the same mechanism is required for AMPAR trafficking, using immunocytochemistry to label surface-expressed GluA2-containing AMPARs. Interestingly, neither Ago2 shRNA nor molecular replacement with S387 mutants had a significant effect on basal levels of surface GluA2, suggesting that GluA2 is not regulated by phosphorylation of Ago2 at S387 under basal conditions (Fig EV5A). NMDAR stimulation caused a significant loss of surface AMPARs, analysed at 20 min after stimulation, which was similar in all transfection conditions, indicating that NMDA-induced AMPAR internalisation is not regulated by phosphorylation at S387. We also analysed total levels of AMPAR subunits GluA1 and GluA2 at 0, 10, 20 and 40 min after NMDAR stimulation. GluA1 has previously been shown to be translationally repressed by miR501-3p in an NMDAR-dependent manner (Hu *et al*, 2015), while a miRNA-dependent regulation of GluA2 translation in response to NMDAR stimulation has not, to our knowledge, been reported. In contrast to LIMK1, expression levels of GluA1 and GluA2 were not rapidly down-regulated at 10 min. While GluA1 showed a significant reduction in expression at 40 min after stimulation, GluA2 expression did not change (Fig EV5B). Furthermore, Akt inhibition had no effect on the NMDA-induced decrease in GluA1 expression (Fig EV5C).

These results indicate that neither NMDAR-stimulated AMPAR internalisation nor modulation of AMPAR subunit expression is controlled by Akt-dependent S387 phosphorylation of Ago2.

### Phospho-regulation of Ago2 at S387 is not required for CA3-CA1 LTD

To investigate the role of Ago2 phosphorylation in the context of synaptic physiology, we analysed basal synaptic transmission and LTD at CA3-CA1 synapses in organotypic hippocampal slices. We used a gene gun to transfect cells with Ago2 shRNA or molecular replacement plasmids. To analyse effects on basal synaptic transmission, we recorded AMPAR EPSCs from transfected (fluorescent) CA1 pyramidal cells and neighbouring untransfected cells in response to the same synaptic stimulus. Ago2 knockdown by shRNA did not significantly alter EPSC amplitude; however, molecular replacement with GFP-S387A-Ago2 caused a significant increase in EPSC amplitude, while GFP-S387D-Ago2 caused a significant decrease (Fig 8A–H).

To directly explore the role of Ago2 phosphorylation in synaptic plasticity, we carried out recordings from CA1 pyramidal cells, and

---

**Figure 7. NMDA-induced dendritic spine shrinkage requires Akt activation, Ago2 phosphorylation at S387 and miRNA-mediated reduction in LIMK1 expression.**

A   S387 phosphorylation is required for NMDA-induced spine shrinkage. Cortical neurons were co-transfected with mRUBY as a morphological marker, and molecular replacement constructs expressing Ago2 shRNA plus shRNA-resistant GFP-Ago2 (WT, S387A or S387D). Forty minutes after NMDA or vehicle application, cells were fixed, permeabilised and stained with anti-mCherry antibody to amplify the mRUBY signal, from which spine size and density were measured. Graph shows quantification of spine size and spine density; *n* = 4 independent experiments (nine cells per condition). *P < 0.05; **P < 0.01; ***P < 0.001; two-way ANOVA, Bonferroni *post hoc* test. Scale bar = 10 μm. Mean ± SEM.

B   Akt activation is required for NMDA-induced spine shrinkage. Cortical neurons were transfected with GFP as a morphological marker, and Akti-1/2 was applied 20 min before NMDA or vehicle application. Forty minutes after NMDA washout, cells were fixed, and spine size and density were measured. Graph shows quantification of spine size and spine density; *n* = 4 independent experiments (10 cells per condition). *P < 0.05; two-way ANOVA, Bonferroni *post hoc* test. Scale bar = 10 μm. Mean ± SEM.

C   Loss of LIMK1 is required for NMDA-induced spine shrinkage. Cortical neurons were co-transfected with mRUBY as a morphological marker, and myc LIMK1 or empty vector. Forty minutes after NMDA or vehicle application, cells were fixed, permeabilised and stained with anti-myc antibody (green) and anti-mCherry antibody (magenta) to amplify the mRUBY signal, from which spine size and density were measured. Graphs show quantification of spine size (left), spine density (middle) and myc LIMK1 expression (right); *n* = 5 independent experiments (10 cells per condition). *P < 0.05; two-way ANOVA, Bonferroni *post hoc* test. Scale bar = 10 μm. Mean ± SEM.

Source data are available online for this figure.

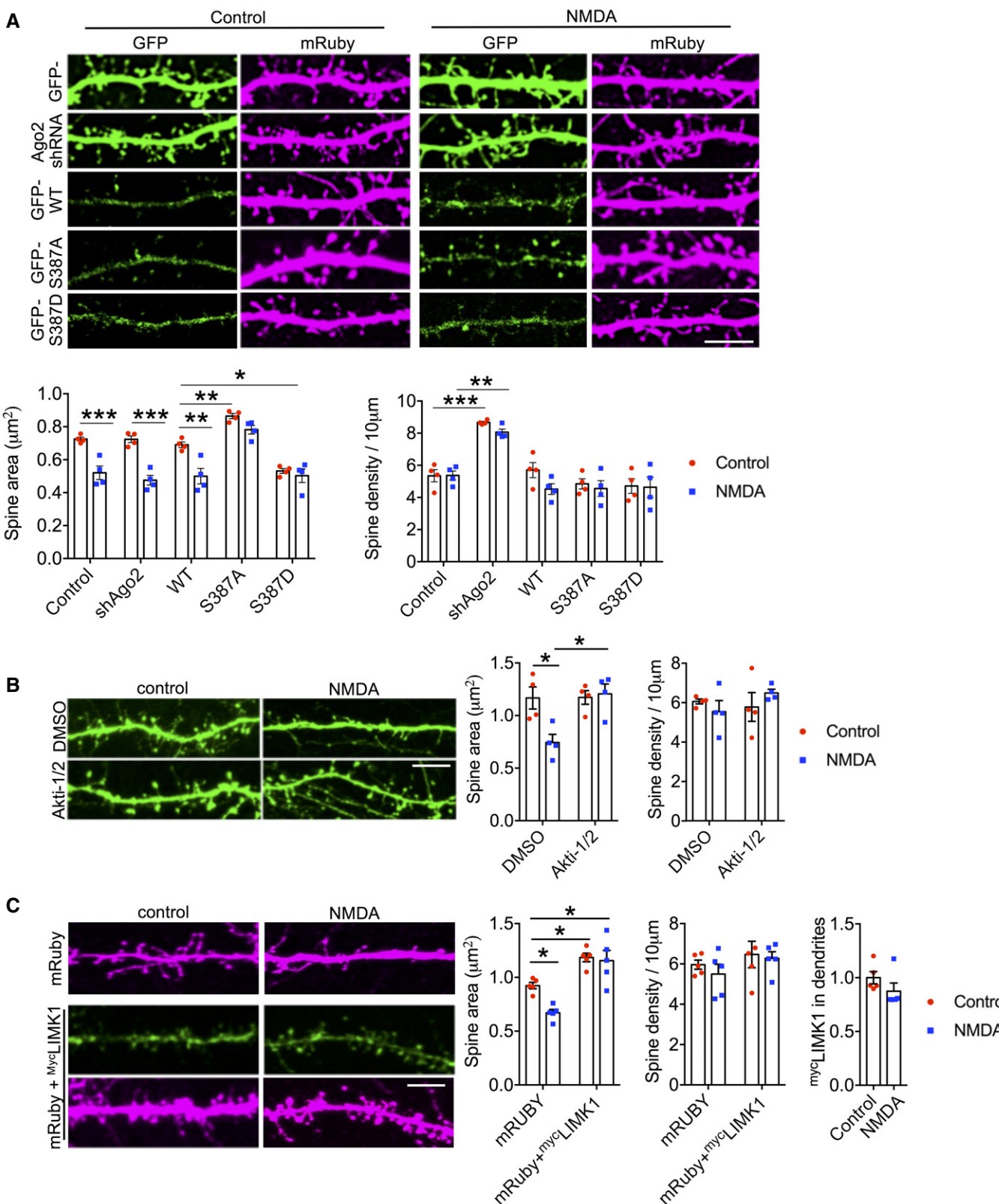

**Figure 7.**

a low-frequency stimulation pairing protocol was used to induce NMDAR-dependent LTD (Rocca *et al*, 2013). Reliable LTD of AMPAR EPSCs was induced in control non-transfected cells but was completely blocked in neurons expressing Ago2 shRNA (Fig 8I and J), indicating that LTD expression requires miRNA activity. LTD was rescued by co-expression of sh-resistant GFP-WT-Ago2, GFP-S387A-Ago2 or GFP-S387D-Ago2 (Fig 8K–N), demonstrating that regulation of Ago2 function by phosphorylation at S387 is not involved in CA3-CA1 LTD. Taken together with our results for AMPAR surface expression, this strongly suggests that, in the context of NMDAR-dependent plasticity, this mechanism is specifically involved in regulating dendritic spine size.

## Discussion

Here, we define a mechanism for rapidly increasing miRNA-mediated translational repression in response to NMDAR stimulation. We show that Ago2 phosphorylation at S387 is transiently increased by NMDAR activation via an Akt-dependent pathway, which enhances the interaction between Ago2 and GW182. Functionally, this mechanism reduces translation of the actin-regulatory protein LIMK1 in neuronal dendrites via miR-134 and has a specific functional role in plasticity. NMDA-induced spine shrinkage is blocked by the Ago2 phospho-null S387A mutant, and a phospho-mimic mutant S387D causes basal spine shrinkage that occludes subsequent plasticity. In contrast, AMPAR internalisation and LTD are unaffected by S387 mutants, suggesting that this is a specific mechanism for structural plasticity.

### Ago2 phosphorylation and protein interactions

Our results strongly suggest that the molecular basis for the modulation of Ago2 function by S387 phosphorylation in neurons is via regulation of GW182 binding. In agreement with our results, a previous study demonstrated that S387 phosphorylation in HeLa cells increased Ago2-GW182 binding and had no effect on miRNA loading (Horman *et al*, 2013). While S387 is situated between the PAZ and MID domains of Ago2, GW182 associates with the PIWI domain towards the C-terminus of Ago2 (Pfaff *et al*, 2013), suggesting that phosphorylated S387 is unlikely to contribute to the GW182 binding site directly. A recent study provides an explanation for this by demonstrating that the LIM-domain protein LIMD1 binds both GW182 and S387-phosphorylated Ago2 in HeLa cells and stabilises the interaction between Ago2 and GW182 in response to Ago2 phosphorylation (Bridge *et al*, 2017). Interestingly, this report further suggests that DDX6 is also recruited to Ago2 by S387 phosphorylation, presumably by interacting with GW182 following LIMD1 binding. In agreement with this observation, our results show that NMDAR stimulation and consequent S387 phosphorylation also enhance Ago2-DDX6 association in neurons. We also show that the interaction of MOV10 with Ago2 is insensitive to NMDAR stimulation and S387 phosphorylation, suggesting only specific Ago2 complexes are regulated in an activity-dependent manner.

While Ago2 contains several phosphorylation sites, there are few reports defining the phospho-regulation of Ago2 in specific physiological contexts, and not all of the proposed phosphorylation sites have been assigned a function. Previous reports have demonstrated that Ago2 phosphorylation at Y529 regulates small miRNA binding (Rudel *et al*, 2011; Mazumder *et al*, 2013), and phosphorylation at Y393 regulates Ago2-Dicer interactions (Shen *et al*, 2013). More recently, phosphorylation of a C-terminal cluster of serine and threonine residues in Ago2 has been shown to regulate mRNA binding (Quévillon Huberdeau *et al*, 2017). Numerous kinases and phosphatases are implicated in synaptic plasticity, and further work is needed to define how various aspects of RISC function might be regulated to control protein translation for plasticity.

---

**Figure 8. CA3-CA1 LTD does not involve phospho-regulation of Ago2 at S387.**

A   Ago2 knock down has no effect on basal synaptic transmission. EPSCs from a neuron transfected with Ago2 shRNA and a neighbouring untransfected control neuron in response to identical stimulation are plotted (*n* = 13 pairs). White dots = individual comparisons, grey dot = mean responses ± SEM. Insets show representative traces. Calibration bars: 20 pA, 50 ms.

B   Quantification of experiments shown in (A). Values are mean ± SEM.

C   Molecular replacement with GFP-WT-Ago2 has no effect on basal synaptic transmission. EPSCs from a neuron transfected with Ago2 shRNA plus GFP-WT-Ago2 and a neighbouring untransfected neuron in response to identical stimulation are plotted (*n* = 13). Black dots = individual comparisons, grey dot = mean responses ± SEM. Calibration bars: 20 pA, 50 ms.

D   Quantification of experiments shown in (C). Values are mean ± SEM.

E   Molecular replacement with GFP-S387A-Ago2 increases basal synaptic transmission. EPSCs from a neuron transfected with Ago2 shRNA plus GFP-S387A-Ago2 and a neighbouring untransfected neuron in response to identical stimulation are plotted (*n* = 13). Black triangles = individual comparisons, grey triangle = mean responses ± SEM. Calibration bars: 20 pA, 40 ms.

F   Quantification of experiments shown in (E). **P < 0.01, paired *t*-test. Error bars represent mean ± SEM.

G   Molecular replacement with GFP-S387D-Ago2 decreases basal synaptic transmission. EPSCs from a neuron transfected with Ago2 shRNA plus GFP-S387D-Ago2 and a neighbouring untransfected neuron in response to identical stimulation are plotted (*n* = 14). Black squares = individual comparisons, grey square = mean responses ± SEM. Calibration bars: 20 pA, 50 ms.

H   Quantification of experiments shown in (G). Values are mean ± SEM. *P < 0.05, paired *t*-test.

I   A pairing protocol (360 pulses at 1 Hz, delivered while the neuron is held at −40 mV) results in LTD in untransfected neurons (*n* = 9). Calibration bars for this and all following example traces: 50 pA, 50 ms. Error bars are mean ± SEM.

J   LTD is blocked in neurons transfected with Ago2 shRNA (*n* = 6). Error bars are mean ± SEM.

K   Co-expression of sh-resistant GFP-WT-Ago2 rescues the block of LTD (*n* = 5). Error bars are mean ± SEM.

L   Co-expression of GFP-S387A-Ago2 rescues the block of LTD (*n* = 5). Error bars are mean ± SEM.

M   Co-expression of GFP-S387D-Ago2 rescues the block of LTD (*n* = 5). Error bars are mean ± SEM.

N   Summary LTD data in panels (I–M). * P < 0.05, one-way ANOVA followed by *post hoc* Bonferroni correction. Error bars are mean ± SEM.

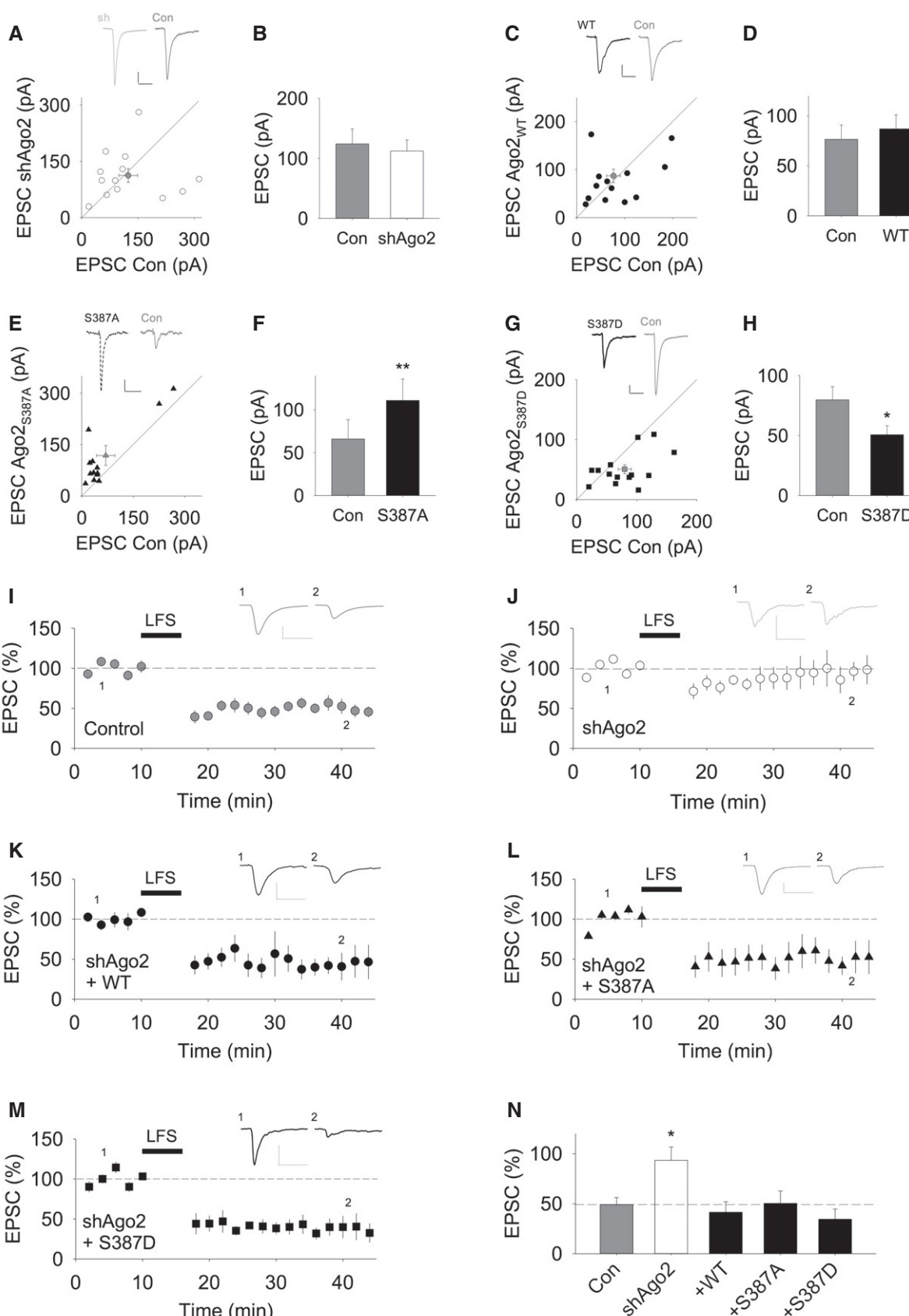

Figure 8.

## NMDAR-dependent miRNA activity

A role for miRNAs in NMDAR-dependent synaptic plasticity has been emerging over the past 5 years, and specific miRNAs have been identified that are required for dendritic spine shrinkage and changes in AMPAR trafficking or expression. For example, a recent study demonstrated that molecular interventions to inhibit the activity of miR-135 block NMDA-induced spine shrinkage at 30 min after stimulation (Hu et al, 2014). While miR-135 (as well as numerous other miRNAs) is up-regulated in response to cLTD induction, transcriptional up-regulation and subsequent processing to mature miRNAs were not detectable until 1 h after stimulation (Hu et al, 2014, 2015). These studies suggest that transcriptional/maturational up-regulation of miRNA expression is not fast enough to play a role in NMDAR-dependent plasticity that is expressed in the first 30 min after stimulation, and is likely to play an important role in later stages of plasticity. A key feature of the mechanism we define here is the speed of translational control. NMDAR stimulation caused significant pS387-dependent silencing of the LIMK1 luciferase reporter via miR-134 at 10 min after stimulation. This timing corresponds to the increase in S387 phosphorylation and increase in Ago2-GW182 interaction, which are both observed by 6 min after stimulation, and return to baseline by 20 min. Therefore, we propose that enhanced RISC activity via NMDAR-dependent Ago2 phosphorylation at S387 can mediate the rapid effect of miRNA during the expression of plasticity by acting on pre-existing miRNAs in the vicinity of the up-regulated RISC complex. We show that this mechanism is not involved in NMDAR-stimulated AMPAR endocytosis, which takes place within 5 min following stimulation (Beattie et al, 2000). Again, this is consistent with the temporal aspects of our results; an increase in RISC activity 6 min after stimulation would not be fast enough to mediate changes in translation to influence such trafficking events.

The NMDAR-stimulated reduction in luciferase activity expressed from the reporter carrying the LIMK1 3′UTR is faster than the reduction in endogenous LIMK1 expression caused by the same stimulus. Indeed, luciferase expression was reduced by ~ 50% after 10 min, whereas endogenous LIMK1 expression took ~ 40 min to reach the same level. This difference is likely to reflect differences in the rate of degradation of luciferase compared to LIMK1. Since Ago2 S387 mutations block or occlude the observed NMDAR-stimulated reduction in luciferase activity and the constructs carrying mutations in the miRNA seed regions are unaffected by NMDAR stimulation, it is unlikely that luciferase degradation is itself stimulated by NMDA and suggests that luciferase has a very rapid turnover time in neurons. While Ago2-GW182 interactions dissociate to baseline levels by 20 min after stimulation, the drop in endogenous LIMK1 expression continues until 40 min. Delays between increased RISC activity and reductions in LIMK1 expression might be explained by the time taken for the degradation of existing protein. Moreover, it is unclear how rapidly translation would return to pre-stimulation levels following dissociation of Ago2-GW182 interactions.

While detectable increases in the endogenous mature forms of specific miRNAs are not seen until 1 h after NMDA stimulation (Hu et al, 2014, 2015), a recent report used an exogenous pre-miRNA fluorescent probe to demonstrate that Dicer activity can be detected in dendrites within a few seconds after stimulation by glutamate uncaging, suggesting that post-transcriptional miRNA maturation can be increased very rapidly (Sambandan et al, 2017). It will be of great interest to determine which miRNAs are processed in this manner, and how this mechanism contributes to the silencing of endogenous proteins during processes such as spine structural plasticity.

Our results show that not all dendritically localised miRNA activities are regulated by this mechanism. While silencing of the luciferase reporter incorporating the LIMK1 3′UTR by miR-134 and of the reporter containing the APT1 3′UTR by miR-138 were both up-regulated by NMDAR stimulation, only LIMK1 silencing by miR-134 was sensitive to S387 mutations in Ago2. Importantly, silencing of the luciferase reporters incorporating the PUM2 or CREB1 3′UTR, both of which are partly mediated by miR-134, is insensitive to S387 mutations. This suggests that a specific, but as yet undefined characteristic of the LIMK1 3′UTR is critical for regulation by S387 phosphorylation.

Our results suggest that LIMK1 and APT1 are targeted in neurons only by miR-134 and miR-138, respectively. In contrast, PUM2 and CREB1 are silenced by miR-134 as well as additional miRNAs that require Ago2 activity. While additional miRNAs have been shown to target LIMK1 in cancer cells (Li et al, 2017), we are not aware of miRNAs that regulate LIMK1 or APT1 in neurons, apart from miR-134 and mir-138, respectively.

## Regulation of dendritic spine size

Our results define a mechanism in which an NMDAR-dependent increase in LIMK1 silencing mediated by Ago2 phosphorylation at S387 in neuronal dendrites causes spine shrinkage as a result of reduced LIMK1 translation and consequent effects on the actin cytoskeleton. In addition, they suggest that complex Ago2-dependent mechanisms are involved in regulating dendritic spine morphogenesis. Basal spine size was unaffected by Ago2 knockdown, but was increased by S387A-Ago2 expression and decreased in neurons expressing S387D-Ago2. Ago2 knockdown caused increases in translation of LIMK1 luciferase reporters and of endogenous LIMK1, suggesting that an alternative Ago family member cannot compensate for the loss of Ago2 in the case of these specific miRNA activities. Spine morphogenesis is influenced by numerous interconnecting signalling pathways, the components of which are translationally controlled by numerous miRNAs. Importantly, the activity of some miRNAs causes spine shrinkage, whereas that of others causes spine growth (Ceman & Saugstad, 2011). For example, miR-134 and miR-138 both promote spine shrinkage, while miR-132 represses Rho GTPase-activating protein p250GAP and promotes dendritic development and spine growth (Bicker et al, 2014). Therefore, we propose that the absence of any detectable effect on spine size caused by Ago2 knockdown can be explained by the attenuation of several silencing events promoting either spine shrinkage or spine growth, with little overall change. While silencing of both LIMK1 and APT1 both promote spine shrinkage, our results show that only LIMK1 silencing is regulated by S387 phosphorylation. Therefore, the spine size phenotype resulting from Ago2 molecular replacement with S387 mutants is likely caused by a specific population of silencing events that includes miR-134-mediated silencing of LIMK1. While this also suggests that Ago2 phosphorylation at S387 does not modulate all silencing events that regulate spine shrinkage, it is possible that this mode of regulation targets only mRNAs involved in this process, via specific features of their 3′UTRs.

Distinct mechanisms appear to regulate basal spine density, which was markedly increased by Ago2 knockdown, but was

insensitive to S387 mutations. This indicates that spine density is regulated by one or more silencing event(s) whose overall effect is to restrict spine density in an Ago2-dependent but pS387-independent manner.

While Akt has been implicated in processes that affect dendritic spine morphology (Kumar *et al*, 2005; Majumdar *et al*, 2011), and NMDAR-stimulated activation of Akt has been shown previously (Perkinton *et al*, 2002; Sutton & Chandler, 2002; but also see Shehata *et al*, 2012), a role for Akt in NMDAR-dependent spine shrinkage has not, to our knowledge, been shown before. Our results identify an Akt-mediated pathway modulating miRNA activity that is specifically involved in controlling dendritic spine size in response to NMDAR stimulation.

### Regulation of synaptic transmission

It has been shown previously that specific miRNAs are required for the maintenance of synaptic transmission as well as synaptic plasticity, including CA3-CA1 LTD (McNeill & Van Vactor, 2012; Hu *et al*, 2014; Hu & Li, 2017). Our results indicate that basal phosphorylation of Ago2 at S387 is involved in restricting EPSC amplitude. This might be a secondary effect caused by modified dendritic spine size in neurons expressing S387 mutants. However, while we do not analyse synaptic GluA2 in our imaging experiments, we show that dendritic GluA2 surface expression is unaffected by the expression of Ago2 mutants. Furthermore, it has been shown previously that LIMK1 depletion does not affect basal synaptic transmission despite altered spine morphology (Meng *et al*, 2002). Therefore, an alternative explanation may be that synaptic transmission is regulated by other miRNA-dependent silencing events that are modulated by Ago2 phosphorylation at S387. Ago2 depletion *per se* had no effect on basal EPSC amplitude, and we propose a similar explanation as described above for basal spine size; some miRNA-mediated silencing events may enhance, while others cause a decrease in EPSC amplitude. Hence, a loss of all Ago2-dependent silencing events may have little net effect.

Our results show that Ago2 knockdown by shRNA blocks LTD, which is consistent with previous work showing that miRNAs are involved in this type of synaptic plasticity (Hu *et al*, 2014). However, LTD does not require phosphorylation of Ago2 at S387, which is in agreement with a previous study demonstrating that LTD is normal in hippocampal slices from *LIMK1* knockout mice (Meng *et al*, 2002). Furthermore, a previous report suggested that Akt activity is not required for the expression of CA3-CA1 LTD (Peineau *et al*, 2009), which is consistent with our observation that Ago2 S387 mutants have no effect on CA3-CA1 LTD.

In agreement with the electrophysiology data, our results show that NMDAR-dependent internalisation of GluA2-containing AMPARs, which is thought to be the major mechanism for LTD (Beattie *et al*, 2000; Anggono & Huganir, 2012), is unaffected by Ago2 S387 mutations, emphasising that distinct pathways control structural and functional plasticity.

In conclusion, we have defined a mechanism for the rapid transduction of NMDAR stimulation into miRNA-mediated translational repression. A particularly intriguing aspect of this mechanism is that Ago2 phosphorylation at S387 regulates some NMDAR-dependent silencing events but not others, suggesting multiple mechanisms are at play in controlling miRNA activity, even in response to the same

stimulus. The RISC machinery involves numerous protein–protein interactions, providing a huge potential for distinct modes of action, and for regulation via a variety of signalling pathways. Future work will determine whether other key protein–protein interactions are subject to regulation by plasticity stimuli or neuromodulators. Numerous neurological disorders involve aberrant miRNA activity (Wang *et al*, 2012; Kocerha *et al*, 2015), and we propose that the mechanism we define here, or similar mechanisms for regulation of RISC activity, might represent targets for therapeutic intervention.

## Materials and Methods

### DNA constructs

Ago2 knockdown and molecular replacement constructs were created by ligating sequences corresponding to shAgo2- and shRNA-resistant Ago2 cDNA into FUGW. shAgo2 oligos: (F: cgcgcccc TGTTCGTGAATTTGGGATCATTGTACAATGATCCCAAATTCACGA ACAtttttaat; R: taaaaaTGTTCGTGAATTTGGGATCATTGTACAATGA TCCCAAATTCACGAACAgggg) were annealed in a thermocycler and cloned into the AscI and PacI sites of FUGW so its expression could be driven by the H1 promoter. shAgo2-resistant Ago2 was created using site-directed mutagenesis (F: TTCAACACAGATCCAT ACGTAAGAGAGTTCGGCATTATGGTCAAAGATGAGATGAC; R: GT CATCTCATCTTTGACCATAATGCCGAACTCTCTTACGTATGGATCT GTGTTGAA) on Myc-Ago2 (kind gift from Dr. Gunter Meister) and then sub-cloned into FUGW (F: TGATCTTGTACAAA ATGTACTC GGGAGCCGGCCC; R: AGATCATGTACATCAAGCAAAGTACATGG TGC) using the BsrGI restriction site on the FUGW vector to create GFP-Ago2. GFP-Ago2 S387A and S387D mutants were created using site-directed mutagenesis S387A F: CAAATTGATGCGAAGTGCAG CTTTCAACACAGATCCATAC; R: GTATGGATCTGTGTTGAAAGCTG CACTTCGCATCAATTTG. S387D F: CAAATTGATGCGAAGTGCAGA TTTCAACACAGATCCATAC; R: GTATGGATCTGTGTTGAAATCTGC ACTTCGCATCAATTTG). mRUBY FUGW was created by replacing the GFP with mRUBY. Luciferase-*LIMK1, APT1, LIN41, CREB1* and *PUM2* 3′UTRs were kindly provided by Prof. G. Schratt.

### Cortical neuronal cultures

Rat embryonic cortical neuronal cultures were prepared using standard procedures from E18 Wistar rat embryos of either gender bred in-house. The culture medium was Neurobasal (Gibco) supplemented with B27 (Gibco) and 2 mM Glutamax. Neurons were plated at densities of 100,000 cells per well of 24-well plate, 500,000 cells per well of a 6-well plate and 900,000 cells per 6-cm dish. Neurons were transfected with plasmid DNA at 10–13 days *in vitro* (DIV) (unless otherwise stated) using Lipofectamine 2000 (Invitrogen) and used for experiments at DIV 15-18. NMDAR stimulation was by bath application of 50 μM NMDA plus 20 μM glycine in HBS (140 mM NaCl, 5 mM KCl, 25 mM HEPES, 1.8 mM CaCl$_2$, 0.8 mM MgCl$_2$ 10 mM glucose, pH 7.4) for 3 min at 37°C and harvested or fixed at specific time points as stated for biochemistry, luciferase assays or imaging. Cells were treated with signalling inhibitors for 20 min prior to NMDA application and were present throughout and after stimulation. Drugs were used at the following concentrations: Akti-1/2 (10 μM), chelerythrine (5 μM), CT99021 (1 μM), KP 372-1

(50 nM), Sc79 (100 nM), Wortmannin (10 μM), rapamycin (50 nM), SB203580 (20 μM). Cultures were incubated with the drugs for 20 min before NMDA stimulation, throughout stimulation and post-stimulation until cells were lysed or fixed.

## Confocal microscopy and image analysis

Cells grown on coverslips were fixed in 4% paraformaldehyde (ThermoFisher) in PBS (Sigma) supplemented with 2% sucrose at RT. Cells were permeabilised in 0.5% NP-40 for 2 min. Coverslips were blocked in 3% BSA (Sigma) for 1 h and incubated with anti-Ago2 (kind gift from Dr. Marvin Fritzler, dilution 1:20), anti-GW182 (kind gift from Dr. Marvin Fritzler dilution 1:100), anti-LIMK1 (Cell Signaling 3842 dilution 1:100), anti-APT1 (Abcam ab91606 dilution 1:200), anti-mCherry (2B Scientific AB0081 dilution 1:400), anti-GFP (NeuroMab N86/8 dilution 1:200) in 3% BSA for 1 h at RT followed by incubation with the appropriate secondary antibodies (Alexa-Fluor 488, 568, 647 ThermoFisher) for 45 min and mounted on slides using mounting medium (Sigma). Coverslips were imaged on a Leica SP5 confocal system under a 63×/1.4 NA oil-immersion or 40×/1.25 NA oil-immersion objectives. The Leica application suite software was used to acquire 0.37-μm stepped Z-stacks throughout the depth of the cells. Image processing and co-localisation analyses were performed using ImageJ, with the experimenter blinded to the experimental condition. At least three independent experiments were performed, and statistical significance was determined using either t-tests or ANOVA, as appropriate. For quantifying co-localisation, the Coloc2 plug-in was used to obtain Pearson's co-localisation co-efficient on three randomly selected dendrites per neuron and the average was used for analysis. Spine sizes and density were measured along 2 dendrites of a neuron with lengths of greater than 80 μm using mRUBY or GFP signal as the morphological marker. Levels of Ago2, LIMK1, APT1 and GluA2 in dendrites were quantified by measuring integrated densities of clusters in at least 2 dendrites per neuron. All error bars on graphs represent standard error of the mean.

## Co-immunoprecipitations

Cells were lysed in IP buffer (0.5% Triton X-100, 150 mM NaCl, 20 mM HEPES, pH 7.4), and 1% of extract was removed for input. The remaining lysate was pre-cleared with protein G-sepharose beads (GE Healthcare) at 4°C for 1 h. 400 μg of cell lysate was incubated with 2 μg of anti-Ago2 antibodies (Cell Signaling 2897) or control IgG (Millipore 12–370) and pulled down with protein G-sepharose beads (GE Healthcare) at 4°C for 1 h. The vehicle control sample was used for the IgG control. Beads were washed three times (1 min each) in 1 ml IP buffer at 4°C. Bound proteins were detected by Western blotting.

## GFP-trap

For GFP-trap (Chromotek) pull-downs, 6 wells of a 6-well plate of DIV 10 cortical neurons were transfected with FUGW-GFP-Ago2 constructs and harvested 5 days post-transfection. GFP-trap was performed according to the manufacturer's instructions. Briefly, cells were lysed in 500 μl lysis buffer (10 mM Tris pH 7.5; 150 mM NaCl; 0.5 mM EDTA; 0.5% NP-40) on ice. 1% was taken as input,

and the remaining extract was incubated with 50 μl of GFP-trap beads for 1 h at 4°C. The beads were then washed with 500 μl of wash buffer (10 mM Tris pH 7.5; 150 mM NaCl; 0.5 mM EDTA) three times (1 min each) at 4°C and bound proteins detected by Western blotting.

## Western blotting

Whole-cell lysates or bound proteins from binding experiments were resolved by SDS–PAGE and transferred to PVDF using a wet transfer apparatus and blocked in 5% milk solution or 5% BSA made up in PBS–Tween. The membranes were blotted with the appropriate primary and secondary antibodies (see below) and bands were visualised using ECL Western blotting substrates (Thermo Fisher Scientific or GE Healthcare). Where appropriate, the membrane was stripped with Restore Western Blot Stripping Buffer (ThermoFisher) and re-probed. Membranes were incubated with the following primary antibodies overnight at 4°C: anti-Ago2 (Cell Signaling Technology clone C34C6 dilution 1:1,000), anti-Ago2 (kind gift from Dr. Marvin Fritzler dilution 1:500), anti-Ago2pS387 (ECM Biosciences AP5291 dilution 1:1,000), anti-GW182 (Novus Bio NBP1-57134 dilution 1:1,000), anti-MOV10 (Abcam ab176687 dilution 1:1,000), anti-DDX6 (Abcam ab45869 dilution 1:1,000), anti-LIMK1 (Cell Signaling 3842 dilution 1:1,000), anti-pan Akt (Cell Signaling 2920 dilution 1:2,000), anti-Akt pS473 (Cell Signaling 4060 dilution 1:2,000), anti-GluA2 (Synaptic Systems #182103 dilution 1:2,000), anti-GAPDH (clone 6C5 dilution 1:20,000), anti-GFP (NeuroMab N86/8 dilution 1:2,000), anti-APT1 (Abcam ab91606 dilution 1:1,000), anti-GluA1 (Millipore AB1504 dilution 1:1,000). Secondary antibodies conjugated to HRP were from GE Healthcare and used at 1:10,000 dilutions for 45 min at RT. For densitometry, Western blot films were scanned and analysed in ImageJ followed by the appropriate statistical analysis. The integrated densities of bands of interest were normalised to appropriate loading controls from the same gel. For pull-downs and co-IPs, bound proteins were normalised to their respective inputs. For GFP-trap experiments, bound proteins were normalised to the GFP-Ago2 signal detected by anti-GFP in the pull-down. All error bars on graphs represent standard error of the mean.

## Luciferase assays

DIV12 cortical cultures were co-transfected with the appropriate luciferase constructs and FUGW-GFP vectors. Dual-luciferase reporter assay system (Promega) was used to perform the assays according to the manufacturer's instructions. Values were normalised for each of at least three independent experiments, and the appropriate statistical analysis was performed. All error bars on graphs represent standard error of the mean.

## Puro-PLA assays

Cortical neurons were treated with 1uM puromycin during bath application in the presence or absence of 50 μM NMDA plus 20 μM glycine in HBS for 3 min at 37°C. Post-stimulation, cells were incubated in conditioned media supplemented with 1 μM puromycin for 40 min. Neurons were then fixed in 4% paraformaldehyde, 2% sucrose at RT for 10 min. Puro-PLA was performed using the

Duolink *in situ* red PLA mouse/rabbit kit (Sigma) according to the manufacturers' protocol. Anti-puromycin (Millipore clone 12D10) and anti-LIMK1 (Cell Signaling 3842) were used at 1:100 dilution. Images were acquired as described above. The number of PLA-positive particles/100 μ of dendrite was quantified as shown in the figures.

### Surface labelling

Cells grown on coverslips were live labelled with anti-GluA2 (Millipore MAB397) diluted 1:30 in HBS for 15 min at RT. Cells were washed three times in HBS and fixed immediately in 4% paraformaldehyde, 2% sucrose at RT for 10 min. Next, the cells were blocked in 3% BSA for 1 h at RT followed by incubation with the appropriate secondary antibody before being mounted. Images were acquired from the coverslips and analysed as described above.

### Organotypic hippocampal slice preparation and biolistic transfection

Organotypic slices were prepared as described previously (Rocca *et al*, 2013). In brief, P7 Wistar rats were sacrificed by cervical dislocation, and the brains were removed and placed in ice-cold cutting solution comprised of 238 mM Sucrose, 2.5 mM KCl, 26 mM $NaHCO_3$, 1 mM $NaH_2PO_4$, 5 mM $MgCl_2$, 11 mM D-glucose and 1 mM $CaCl_2$. Transverse hippocampal slices (350 μm) were cut using a Leica VT122 S vibratome, washed three times in culture media and plated on Millicell culture plate inserts (Millipore Corporation, Bedford, MA, USA) in 6-well plates containing culture medium. Culture medium comprised 78.8% minimum essential medium, 20% heat-inactivated horse serum, 30 mM HEPES, 16 mM D-glucose, 5 mM $NaHCO_3$, 1 mM $CaCl_2$, 2 mM $MgSO_4$, 68 μM ascorbic acid, 1 μg/ml insulin, pH adjusted to 7.3 and 320–330 mOsm. The slices were then cultured in an incubator (35°C, 5% $CO_2$) for 6–11 days *in vitro* (DIV) before biolistic transfection with gene gun bullets prepared as described previously (O'Brien & Lummis, 2006). Electrophysiological recordings were made from slices from 2 to 5 days post-transfection.

### Electrophysiology

Whole-cell patch-clamp electrophysiology experiments were performed on transfected cells, visualised using fluorescence microscopy, and in some cases neighbouring untransfected cells. Recordings were performed in ACSF comprised of 119 mM NaCl, 2.5 mM KCl, 26 mM $NaHCO_3$, 1 mM $NaH_2PO_4$, 4 mM $CaCl_2$, 4 mM $MgCl_2$, 11 mM D-glucose, 0.05 mM picrotoxin and 0.001–0.01 mM 2-chloroadenosine (bubbled with 95% $O_2$/5% $CO_2$). Stimulating electrodes were placed in the Schaffer collateral pathway, and pyramidal neurons in area CA1 were voltage-clamped at −60 mV using pipettes with resistance 3–4 MΩs fabricated using a Sutter P-97 micropipette puller (Sutter Instruments, CA, USA). Pipettes contained solution comprised of 130 mM $CsMeSO_4$, 8 mM NaCl, 4 mM Mg-ATP, 0.3 mM Na-GTP, 0.5 mM EGTA, 10 mM HEPES, 6 mM QX-314 (pH 7.25, 290 mOsm). Recordings were made using an Axon Instruments Multiclamp 700A or 700B (Molecular Devices, Berkshire, UK). Excitatory post-synaptic currents (EPSC) amplitude, series resistance, input resistance and DC were monitored and analysed online and offline using the WinLTP software (Anderson & Collingridge, 2007). Only cells with series resistance < 30 MΩ with a change in series resistance < 20% from the start were included in this study. Statistical significance was tested using a one-way ANOVA followed by a *post hoc* Bonferroni's multiple comparison test in SigmaPlot 13 (Systat Software Inc., San Jose, CA, USA).

**Expanded View** for this article is available online.

### Acknowledgements

We thank P. Rubin and Y. Nakamura for technical assistance, Prof. G. Meister for Myc-Ago2, Prof. D. Stephens for mRUBY, Prof. G. Schratt for luciferase-*LIMK1, APT1, LIN41, PUM2 and CREB1* 3′UTRs and Prof. G. Thomas for Myc-LIMK1. Anti-GW182 was kindly donated by Prof. M. Fritzler. We thank Dr. Z. Koszegi, Dr. Y. Nakamura and Prof. J. Uney for critical reading of the manuscript. All imaging experiments were carried out in the Wolfson Bioimaging Facility at the University of Bristol. This work was funded by BBSRC and MRC.

### Author contributions

DR designed and performed biochemistry and imaging experiments; TMS and MA designed and performed electrophysiology experiments; GLC supervised electrophysiology experiments; JGH designed and supervised the experiments and supervised the project. DR and JGH wrote the paper.

### Conflict of interest

The authors declare that they have no conflict of interest.

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
