## [Review Process File · The EMBO Journal]

NMDAR-dependent Argonaute 2 phosphorylation regulates miRNA activity and dendritic spine plasticity

Dipen Rajgor, Thomas M Sanderson, Mascia Amici, Graham Collingridge, Jonathan G Hanley

Review timeline:

Submission date:	7 August 2017
Editorial Decision:	18 September 2017
Revision received:	12 February 2018
Editorial Decision:	19 March 2018
Revision received:	27 March 2018
Accepted:	29 March 2018

Editor: Karin Dumstrei

Transaction Report:

1st Editorial Decision

18 September 2017

Thank you for submitting your manuscript to The EMBO Journal. Your study has now been seen by three referees and their comments are provided below.

As you can see from the comments, the referees find the analysis interesting and insightful. They bring up a number of valid points that I anticipate you should be able to sort out in a good manner. I would therefore like to invite you to submit a revised version of the manuscript, addressing the comments of all three reviewers. I should add that it is EMBO Journal policy to allow only a single major round of revision and that it is therefore important to address the raised concerns at this stage.

When preparing your letter of response to the referees' comments, please bear in mind that this will form part of the Review Process File, and will therefore be available online to the community. For more details on our Transparent Editorial Process, please visit our website:
http://emboj.embopress.org/about#Transparent_Process

Thank you for the opportunity to consider your work for publication. I look forward to your revision.

REFeree REPORTS

Referee #1:

In the present manuscript, Rajgor and Hanley explore the mechanisms involved in NMDAR-dependent dendritic spine plasticity in primary neurons. They report a novel mechanism whereby brief NMDAR stimulation rapidly induces phosphorylation of the core RISC factor Ago2 via activation of the Akt signalling pathway, which in turn causes increased association of Ago2 with another RISC factor, GW182, and increased miRNA-mediated translational repression of the miR-134 target gene LIMK1. This pathway is further shown to be required for NMDA-mediated spine shrinkage.

miRNAs have been previously implicated in NMDAR-dependent synaptic plasticity, but the mechanisms underlying alterations in miRNA activity upon NMDA stimulation have not been characterized. Therefore, this study adds to our understanding of miRNA function in synaptic plasticity and is in principle suitable for publication in the EMBO Journal.

Overall, the conclusions drawn by the authors are supported by the presented datasets. However, several issues need to be addressed before publication. In particular, the authors have to demonstrate that NMDA treatment indeed results in a very rapid repression of miRNA-dependent protein synthesis of endogenous target genes, such as LIMK1, and that this repression is indeed required for NMDA function. In addition, the presentation of experimental statistics has to be improved.

Major concerns:

- As shown in Fig. 4, NMDA increases Ago/GW182 association and Ago2 phosphorylation within a few minutes. Similarly, spine shrinkage is observed already 10min after NMDA washout. It is rather astonishing that within this time frame, concentrations of key effector proteins (e.g. LIMK1) should be already significantly altered as a result of increased miRNA-dependent translational repression (rather than increased turnover of pre-existing protein). Therefore, the authors have to investigate the rate of de novo (!) protein synthesis of proteins such as LIMK1 preferentially in dendrites.
- Levels of Limk1 drop continuously over a period of 40min (Fig. 6A), whereas increases in Ago phosphorylation and Ago/GW182 association are very transient (Fig. 4). Given this discrepancy, it is possible that reductions in Limk1 levels are not causally involved in NMDA-mediated spine shrinkage, but rather a by-product. The authors have to assess whether re-expression of LIMK1 (or other miRNA targets) is indeed sufficient to prevent spine shrinkage in the context of NMDA treatment.
- In a number of cases, the statistical description of experiments was rather superficial and sometimes questionable. For example, in Ago/GW182 co-localization experiments (Fig. 1C,D, Fig. 3B), the total number of cells from several experiments is used as N for the statistics, but it is not stated how many cells originate from each individual experiment. It would be better practice to consider independent experiments as N, rather than individual cells (which cannot be treated equally since they come from different experiments). The same holds true for GluA2 and spine data in Fig. 7 and 8, respectively. In Fig. 5, it is not stated what N refers to (independent experiments? Replicate measurements?).

More minor concerns:

- In Fig. 2C, the band intensities in the blot (lanes sc79) do not correspond to the quantification presented on the right
- In Fig. 5, it would be important to show also the comparison between wt and mut reporter constructs in order to assess the effect of the endogenous miRNA on target gene repression under basal conditions, in particular with respect to the surprisingly strong effect of Ago2 knockdown on the Limk1 and Apt1 reporters. It is also rather surprising that mutant reporters do not react to Ago2 knockdown at all, given that they presumably contain other miRNA binding sites in addition to the ones mutated. Again, the strong effect of Ago2 knockdown would favour the idea that multiple miRNA sites are functional, so there is some inconsistency here. These issues have to be at least discussed.
- In Fig. 7, it is rather surprising that only spine density is sensitive to Ago2 knockdown, given previous publications that report a role for microRNAs miR-134 and miR-138 in the control of spine size, but not spine density. This has to be discussed.
- Fig. 8 reports mostly negative data and could be moved into the supplement.

Referee #2:

In this manuscript, Rajgor and Hanley reported a rapid and transient, Akt-mediated Ago2-

phosphorylation (S387) event in primary neuronal cultures that facilitates translational repression of LIMK1 in the context of NMDA-triggered chemical LTD. The same group has recently reported a NMDA-dependent Ca²⁺-sensitive dissociation mechanism of Ago2 from endosomal PICK1, also resulting in activation of microRNA-mediated translation repression (Rajgor et al., 2017). The current study is novel. The novelty lies in that NMDA stimulation results in Akt-mediated S387 phosphorylation of Ago2; this modification causes translation repression of specific mRNA (LIMK1 but not APT1), and induces specific synaptic phenotypes (blocks spine head shrinkage during cLTD but not AMPAR trafficking). The findings are important for linking synaptic activities to local translation regulations and synaptic plasticity.

Major points:

1. A critical but unanswered question in this manuscript is regarding the physiological role of this particular NMDA-Akt-Ago2 S387-LIMK1 signaling axis in cLTD (given selective translation repression and phenotype). Experiments to address this question will be to measure cLTD in the molecular replacement experiments using S387A and S387D constructs.
2. Recent publication by the same group Rajgor et al., 2017 also focused on the cellular mechanisms to activate Ago2-microRNA mediated translation repression pathways but found distinct mechanisms there. In Rajgor et al., 2017, the authors reported a mechanism that involves dissociation of Ago2 from endosomal PICK1 in a Ca²⁺-dependent manner; while the new mechanism found in this study operates by Akt-mediated phosphorylation of Ago2 resulting in Ago2 interaction with GW182 and DDx6. Since both pathways are triggered by NMDA, it is important for the authors to provide perspectives on how these two pathways may interact in cells at basal state and after stimulation. Although the authors demonstrated using GFP-Trap that S387 phosphorylation does not affect NMDAR-dependent GFP-Ago2 dissociation from PICK1, these experiments do not tell us whether dissociation from PICK1 occurs prior to, and is required for Ago2 phosphorylation. Overexpressing PICK1-AX10 in the molecular replacement will address this point.
3. Quantifications in this manuscript can be improved by replacing the bar graphs with box-whisker plots (alternatively if there are not many data points, a simple presentation of the data points).
4. Figure 2C, WB image of GW182 is not visually supporting quantification bar graphs to the right (especially in the sc79 lanes). Given that this is a critical piece of finding in the current study, a more representative image should be used.

Minor points:

1. Figure 3, label with "DDx6" instead of "p54";
2. In Figure 6, APT1 staining images will be appreciated. However, if a good antibody allowing detection of APT1 in dendrites is not available, the reviewer understands.
3. Quantification of the WB bands using Image J is key to interpreting the results. It will help the readers better estimate the differences if the authors provided more details in methods section on this quantification procedure. For example, is integrated density or average intensity used? are data points representing normalized intensity to loading control on the same blot?
4. Mutagenesis in constructing resistant WT Ago2 contains 4 additional mutations outside of the targeted sites in addition to the 8 mutations in the targeted region. What are the rationales for making additional mutations in the non-targeted regions?

Referee #3:

Rajgor and Hanley

NMDAR-dependent Argonaute 2 phosphorylation regulates miRNA activity and dendrite spine plasticity

In this manuscript, the authors have analyzed Ago2 phosphorylation at S387 in dendritic spines of cultured neurons. In co-IPs and co-localization experiments, they find that Ago2 - GW182 interactions are enhanced upon NMDAR stimulation. It has been reported before that Akt-mediated phosphorylation of Ago2 at S387 enhances the interaction with GW182. Consequently, the authors report that this phosphorylation is required for the observed NMDAR effects. The miRNA that is predominantly affected by this mechanism appears to be miR-134, which is regulated by Ago2 S387 phosphorylation and dependent on NMDAR signaling. It has been shown before that miR-134 regulates Limk1, which is a regulator of spine morphology. Indeed, the authors find that Limk1 protein expression is reduced upon NMDA stimulation. Consistently with all data presented, NMDA stimulation leads to Ago2 S387 phosphorylation, increased miR-134 activity and thus reduction of Limk1 levels. This results in dendritic spine shrinkage. MiR-138 and its target APT, however, seem to be independent of this pathway. Finally, they report that Ago2 S387 phosphorylation is not important for NMDAR-stimulated AMPAR trafficking, which is important for long-term potentiation.

This is an interesting and clear analysis of Ago2 S387 phosphorylation in neurons. The manuscript is well written and the results are presented in a very clear way. It adds novel aspects to Ago regulation as well as dendritic spine function. Most of the effects that are observed are rather mild, but I understand that is due to the neuronal system, where such effects are standard and therefore not generally problematic. I have a few comments and/or suggestions that are listed below.

1. The authors analyzed GW182 in their assays. However, in mammals, three GW proteins exist (TNRC6A-C). It is unclear which specific GW protein is recognized by their antibody. In the current model, the three TNRC6 proteins might function similarly and therefore it would be nice to see how binding of all three proteins is affected by Ago2 S387 phosphorylation.
2. Similarly, here only Ago2 is analyzed. However, there are at least two more Ago proteins (Ago1 and Ago3) expressed in these cells. It is believed that Ago proteins could function somewhat redundantly and therefore such clear Ago2-specific effects are probably unexpected. It would be nice to analyze the other Agos as well. Alternatively, upon Ago2 knock down, rescue with Ago1 or Ago3 could be performed. Maybe a simple increase in these proteins is already sufficient. That would also serve as another control.
3. Figure 5: loading controls of the rescues should be included. Although the mutants seem to express at similar levels, each individual experiment should include such a control.
4. The authors claim that the activity of miR-134 is affected by NMDAR signaling and subsequent Ago2 phosphorylation, which would be not so easy to explain mechanistically. In an alternative model, there could be some kind of unknown changes on the Limk1 mRNA that makes it more accessible to Ago2 pS387. The authors could/should easily test that. They could use a reporter construct that does not contain the Limk1 3' UTR but only a miR-134 binding site. If this is also affected, it could be claimed that miR-134 activity is affected. If not, it might be caused by the Limk1 3' UTR itself (e.g. structural changes, RBP, which would make the target site more accessible etc.). In addition, other miR-134 targets could be tested if they are also affected.

1st Revision - authors' response

12 February 2018

Referee #1:

Major concerns:

- As shown in Fig. 4, NMDA increases Ago/GW182 association and Ago2 phosphorylation within a few minutes. Similarly, spine shrinkage is observed already 10min after NMDA washout. It is rather astonishing that within this time frame, concentrations of key effector proteins (e.g. LIMK1) should be already significantly altered as a result of increased miRNA-dependent translational repression (rather than increased turnover of pre-existing protein). Therefore, the authors have to investigate the rate of de novo (!) protein synthesis of proteins such as LIMK1 preferentially in dendrites.

The data presented in figure 7A demonstrate spine shrinkage at 40 min after stimulation; we did not analyse a 10 min time point. We demonstrated previously that significant spine shrinkage takes ~30-40 min after NMDAR stimulation (Nakamura et al., 2011), and we measured LIMK1 synthesis in dendrites over this time frame using Puro-PLA assays. Using this method, we observed that NMDAR stimulation caused a significant reduction in the number of sites of LIMK1 translation initiated within 40 min after stimulation. This demonstrates that LIMK1 synthesis in dendrites is regulated in an NMDAR-dependent manner within 40 min. Unfortunately, we were unable to detect PLA puncta at earlier time points, so we could not analyse the effect of NMDA at earlier time points. A possible explanation for this is the C-terminal epitope recognised by the LIMK1 antibody, which results in less efficient PLA compared to N-terminal epitopes (tom Dieck et al 2015). This was the most effective antibody available to us for these experiments, and we were able to detect robust and specific Puro-PLA signal at the 40 min time point.

To demonstrate that the NMDAR-dependent reduction in sites of nascent LIMK1 translation is caused by Ago2 phosphorylation at S387, we performed Puro-PLA for nascent LIMK1 in neurons expressing Ago2 shRNA. Neurons expressing Ago2 shRNA alone showed a significant increase in dendritic PLA puncta, which was rescued by co-expression of sh-resistant Ago2. In neurons expressing Ago2 shRNA alone, NMDAR stimulation had no effect on PLA puncta. This confirms that dendritic LIMK1 synthesis is reduced by NMDAR stimulation in a miRNA-dependent manner. Molecular replacement with Ago2 S387A completely blocked the NMDAR-dependent reduction in sites of nascent LIMK1 synthesis, strongly suggesting that S387 phosphorylation is required. Moreover, Ago2 S387D expression mimicked and occluded the effect of NMDAR stimulation. Together, these experiments show that NMDAR stimulation causes a decrease in LIMK1 translation in dendrites in a miRNA-dependent manner that requires Ago2 S387 phosphorylation. These new results are presented in Fig.6D of the revised manuscript.

- Levels of Limk1 drop continuously over a period of 40min (Fig. 6A), whereas increases in Ago phosphorylation and Ago/GW182 association are very transient (Fig. 4). Given this discrepancy, it is possible that reductions in Limk1 levels are not causally involved in NMDA-mediated spine shrinkage, but rather a by-product. The authors have to assess whether re-expression of LIMK1 (or other miRNA targets) is indeed sufficient to prevent spine shrinkage in the context of NMDA treatment.

We have performed this experiment as suggested. We used a myc-tagged LIMK1 construct that does not contain the native LIMK1 3'UTR, and is therefore resistant to regulation by miR-134. Neurons expressing myc-LIMK1 are resistant to NMDA-induced spine shrinkage, which strongly suggests that a reduction in LIMK1 expression is required for spine shrinkage. These data are consistent with findings from another group (Calabrese et al., 2014). This experiment is presented in Fig 7C of the revised manuscript.

While Ago2-GW182 interactions return to baseline by 20 min after stimulation, it is unclear how rapidly after this time point translation would return to pre-stimulation levels. Moreover, delays between increased RISC activity and reductions in LIMK1 expression can be explained by the time taken for the degradation of existing protein. This is discussed on P.17 of the revised manuscript.

- In a number of cases, the statistical description of experiments was rather superficial and sometimes questionable. For example, in Ago/GW182 co-localization experiments (Fig. 1C,D, Fig. 3B), the total number of cells from several experiments is used as N for the statistics, but it is not stated how many cells originate from each individual experiment. It would be better practice to consider independent experiments as N, rather than individual cells (which cannot be treated equally since they come from different experiments). The same holds true for GluA2 and spine data in Fig. 7 and 8, respectively. In Fig. 5, it is not stated what N refers to (independent experiments? Replicate measurements?).

We apologize for the lack of details included in the figure legends for some of the experiments. We have clarified this in the revised manuscript. Moreover, we have reanalysed the data from the experiments presented in Figs 1C,1E, 3B, 6C, 6D, 7A, 7B, 7C, Figure EV5 considering each independent experiment as N.

More minor concerns:

- In Fig. 2C, the band intensities in the blot (lanes sc79) do not correspond to the quantification presented on the right

We thank the reviewer for pointing this out. We have replaced this blot with an example that is more representative of the quantified data.

- In Fig. 5, it would be important to show also the comparison between wt and mut reporter constructs in order to assess the effect of the endogenous miRNA on target gene repression under basal conditions, in particular with respect to the surprisingly strong effect of Ago2 knockdown on the Limk1 and Apt1 reporters. It is also rather surprising that mutant reporters do not react to Ago2 knockdown at all, given that they presumably contain other miRNA binding sites in addition to the ones mutated. Again, the strong effect of Ago2 knockdown would favour the idea that multiple miRNA sites are functional, so there is some inconsistency here. These issues have to be at least discussed.

We thank the reviewer for this suggestion. We have presented a comparison between all WT and mutant reporter constructs under basal conditions in Appendix Figure S4 of the revised manuscript. Surprisingly, the mutant constructs for *LIMK1*, *APT1* and *LIN41* showed similar expression in control and shAgo2 neurons, suggesting that miR-134, miR-138 and Let-7 respectively are the only miRNAs with activity on these 3'UTRs in neurons. In contrast, while the *CREB1* and *PUM2* constructs with mutated miR-134 seed region showed less repression compared to WT constructs, repression was reduced further in shAgo2 neurons. This suggests that *CREB1* and *PUM2* 3'UTRs contain functional binding sites for additional miRNAs. This is discussed on P.9 and P.17 of the revised manuscript.

- In Fig. 7, it is rather surprising that only spine density is sensitive to Ago2 knockdown, given previous publications that report a role for microRNAs miR-134 and miR-138 in the control of spine size, but not spine density. This has to be discussed.

We included a discussion about this issue on PP 15-16 of the original version of the manuscript (P.18 of the revised manuscript). Our results with the Ago2 molecular replacement constructs indicate that spine size is affected by miRNA activity. The lack of effect of Ago2 shRNA alone could be explained by the attenuation of numerous silencing events that individually promote either spine shrinkage or spine growth, but together produce little overall change.

- Fig. 8 reports mostly negative data and could be moved into the supplement.

We have moved these data to Figure EV5.

Referee #2:

Major points:

1. A critical but unanswered question in this manuscript is regarding the physiological role of this particular NMDA-Akt-Ago2 S387-LIMK1 signaling axis in cLTD (given selective translation repression and phenotype). Experiments to address this question will be to measure cLTD in the molecular replacement experiments using S387A and S387D constructs.

In collaboration with Graham Collingridge's lab, we have investigated the physiological role of the novel mechanism in hippocampal LTD. Rather than testing chemical LTD, we chose to analyse CA3-CA1 LTD in organotypic hippocampal slices using a low frequency stimulation protocol, which is an NMDAR-dependent form of plasticity, and is generally accepted to be more physiologically relevant. We used a gene-gun to transfect CA1 neurons in the slice with the same molecular replacement constructs that we used for our cell biology experiments. LTD was completely abolished by Ago2 knockdown and fully rescued by expression of sh-resistant WT-Ago2, demonstrating that Ago2 is required for LTD. Interestingly, sh-resistant S387A-Ago2 and S387D-Ago2 also rescued the LTD phenotype, indicating that S387 phosphorylation is not involved in LTD of CA3-CA1 synaptic transmission.

Nonetheless, we found that S387A-Ago2 expression caused a significant increase in basal synaptic transmission, and S387D-Ago2 caused a corresponding decrease. Since S387A-Ago2 caused an increase, and S387D caused a decrease in dendritic spine size, perhaps the observed effects on synaptic transmission are caused by changes in spine size. However, it has been shown previously that LIMK1 depletion does not affect basal synaptic transmission despite changes in spine morphology (Meng et al., 2002), therefore an alternative explanation may be that synaptic

transmission is regulated by other miRNA-dependent silencing events that are modulated by Ago2 phosphorylation at S387.

Taken together, our results indicate that the regulation of miRNA activity by NMDAR-dependent S387 phosphorylation is a mechanism that is specific to structural plasticity, and not involved directly in plasticity of synaptic transmission. However, the phosphorylation state of Ago2 at S387 regulates basal synaptic transmission via an as yet undefined mechanism. The electrophysiology data are presented in Fig.8 of the revised manuscript.

2. Recent publication by the same group Rajgor et al., 2017 also focused on the cellular mechanisms to activate Ago2-microRNA mediated translation repression pathways but found distinct mechanisms there. In Rajgor et al., 2017, the authors reported a mechanism that involves dissociation of Ago2 from endosomal PICK1 in a Ca²⁺-dependent manner; while the new mechanism found in this study operates by Akt-mediated phosphorylation of Ago2 resulting in Ago2 interaction with GW182 and DDX6. Since both pathways are triggered by NMDA, it is important for the authors to provide perspectives on how these two pathways may interact in cells at basal state and after stimulation. Although the authors demonstrated using GFP-Trap that S387 phosphorylation does not affect NMDAR-dependent GFP-Ago2 dissociation from PICK1, these experiments do not tell us whether dissociation from PICK1 occurs prior to, and is required for Ago2 phosphorylation. Overexpressing PICK1-AX10 in the molecular replacement will address this point.

We demonstrated previously that PICK1-Ago2 interactions are disrupted immediately after NMDA washout (Rajgor et. al 2017), and in the current study we show that Ago2 phosphorylation at S387 occurs at later time points. To determine whether Ago2-PICK1 dissociation is required for NMDAR-mediated Ago2 phosphorylation, we performed the experiment suggested by the reviewer, and investigated whether molecular replacement of endogenous PICK1 with GFP-PICK1-AX10 (whose interaction with Ago2 is resistant to NMDAR stimulation) affects Ago2 phosphorylation at S387. In the presence of PICK1-AX10, NMDAR stimulation caused a significant increase in pS387 that was indistinguishable from neurons expressing WT-PICK1, demonstrating that dissociation from PICK1 is not a pre-requisite for S387 phosphorylation. Therefore, while S387 phosphorylation occurs after dissociation from PICK1, the latter event does not depend on the former. These new results are presented in Appendix Fig.S2 of the revised manuscript.

3. Quantifications in this manuscript can be improved by replacing the bar graphs with box-whisker plots (alternatively if there are not many data points, a simple presentation of the data points).

We have updated all of the graphs in the revised version of the manuscript to show individual data points.

4. Figure 2C, WB image of GW182 is not visually supporting quantification bar graphs to the right (especially in the sc79 lanes). Given that this is a critical piece of finding in the current study, a more representative image should be used.

We thank the reviewer for pointing this out. We have replaced this blot with an example that is more representative of the quantified data.

Minor points:

1. Figure 3, label with "DDx6" instead of "p54";

We thank the reviewer for pointing this out, and we have changed p54 to DDX6

2. In Figure 6, APT1 staining images will be appreciated. However, if a good antibody allowing detection of APT1 in dendrites is not available, the reviewer understands.

We have performed APT1 staining in neuronal dendrites expressing the Ago2 molecular replacement constructs in resting and NMDAR stimulated neurons. In agreement with our luciferase assays, endogenous APT1 levels decrease to similar levels after NMDAR stimulation in control neurons and in neurons where endogenous Ago2 has been replaced by GFP-Ago2 WT or S387 phospho-mutants. NMDA-induced APT1 downregulation is blocked in neurons where Ago2 has been depleted and these neurons express significantly more dendritic APT1 than control neurons under both resting and stimulated conditions. This experiment is presented in Fig.EV4 of the revised manuscript.

3. *Quantification of the WB bands using Image J is key to interpreting the results. It will help the readers better estimate the differences if the authors provided more details in methods section on this quantification procedure. For example, is integrated density or average intensity used? are data points representing normalized intensity to loading control on the same blot?*

We apologise for the lack of information regarding quantification of Western blot data. Image J was used to calculate the integrated density of each band and was then normalized to an appropriate loading control on the same gel. For co-IPs, bound proteins were normalized to their respective inputs. For GFP-trap experiments, bound proteins were normalized to amount of GFP-PICK1 or GFP-Ago2 detected by anti-GFP. We have now added this to the materials and methods.

4. *Mutagenesis in constructing resistant WT Ago2 contains 4 additional mutations outside of the targeted sites in addition to the 8 mutations in the targeted region. What are the rationales for making additional mutations in the non-targeted regions?*

There is no rationale for this apart from making absolutely sure that the Ago2 expressed from our molecular replacement constructs would be completely shRNA-resistant. All are silent mutations, so there are no changes introduced to the amino acid sequence.

Referee #3:

1. *The authors analyzed GW182 in their assays. However, in mammals, three GW proteins exist (TNRC6A-C). It is unclear which specific GW protein is recognized by their antibody. In the current model, the three TNRC6 proteins might function similarly and therefore it would be nice to see how binding of all three proteins is affected by Ago2 S387 phosphorylation.*

In this study we have analysed the interaction between Ago2 and GW182 (TNRC6A); the epitope for the antibody we used is not present in TNRC6B and C. To our knowledge, suitable antibodies to detect endogenous TNRC6B or TNRC6C via blotting or imaging on primary rat neuronal cultures are not available.

Nevertheless, we have used a heterologous system to investigate whether Ago2 S387 phospho-mutations affect binding to TNRC6A, B and C. We co-transfected Flag-tagged TNRC6 proteins with GFP-Ago2 WT, S387A or S387D in HEK cells and performed GFP trap pull-downs. Interestingly, all 3 GW proteins showed greater binding to S387D and reduced binding to S387A compared to WT, strongly suggesting that they are all regulated in a similar manner by S387 phosphorylation. These data are presented in Figure EV2 of the revised manuscript.

2. *Similarly, here only Ago2 is analyzed. However, there are at least two more Ago proteins (Ago1 and Ago3) expressed in these cells. It is believed that Ago proteins could function somewhat redundantly and therefore such clear Ago2-specific effects are probably unexpected. It would be nice to analyze the other Agos as well. Alternatively, upon Ago2 knock down, rescue with Ago1 or Ago3 could be performed. Maybe a simple increase in these proteins is already sufficient. That would also serve as another control.*

We have performed a number of experiments to address the question of Ago specificity:

1) To investigate whether Ago1 shows NMDAR-dependent increase in GW182 binding, we performed co-immunoprecipitations with anti Ago1 antibody. The Ago1-GW182 and Ago1-DDX6 interactions did not increase after NMDAR stimulation, suggesting that Ago1 RISC complexes are not regulated in an NMDA-dependent manner. We also attempted to perform similar co-IP experiments for Ago3, but unfortunately, we found that the anti-Ago3 antibody was unsuitable for co-IP.

2) To investigate whether Ago2 knockdown causes an upregulation of Ago1 or Ago3 expression, we reduced Ago2 expression by lentivirus-delivered shRNA and analysed levels of Ago1 and Ago3 in these neurons. While Ago2 shRNA caused a very efficient Ago2 knockdown, expression levels of Ago1 and Ago3 were unaffected, suggesting no compensation at the level of Ago protein expression.

3) To investigate whether Ago1 can functionally compensate for Ago2 knockdown in miR134-mediated *LIMK1* silencing, we co-expressed myc-tagged Ago1 with Ago2 shRNA in luciferase assays. Interestingly, ^{myc}Ago1 expression did not reverse the reduced translational repression caused by Ago2 knockdown, strongly suggesting that miR134-mediated *LIMK1* silencing is an Ago2-specific event.

These new results are presented in Figure EV1 and Appendix Figure S3 of the revised manuscript.

3. Figure 5: loading controls of the rescues should be included. Although the mutants seem to express at similar levels, each individual experiment should include such a control.

We acknowledge the referee's concern with this issue, however this type of control is technically challenging and almost impossible to do when performing luciferase assays on neurons, because the transfection efficiency is low. Since most of the lysate is needed for the luciferase assay itself, detection of GFP-Ago2 via Western blotting would be impossible on the small amount remaining. We note that previous studies using similar techniques to knockdown protein expression adopt the same approach (e.g., Storchel et al., 2015).

Furthermore, we have shown that all three GFP-Ago2 constructs express to similar levels (see Appendix Fig S1), and they rescue to similar levels in the *APT1*, *LIN41*, *CREB1* and *PUM2* luciferase assays (see Fig. 5). We are therefore confident that the expression levels are equal between the WT and mutants in the luciferase assays.

4. The authors claim that the activity of miR-134 is affected by NMDAR signaling and subsequent Ago2 phosphorylation, which would be not so easy to explain mechanistically. In an alternative model, there could be some kind of unknown changes on the *Limk1* mRNA that makes it more accessible to Ago2 pS387. The authors could/should easily test that. They could use a reporter construct that does not contain the *Limk1* 3' UTR but only a miR-134 binding site. If this is also affected, it could be claimed that miR-134 activity is affected. If not, it might be caused by the *Limk1* 3' UTR itself (e.g. structural changes, RBP, which would make the target site more accessible etc.). In addition, other miR-134 targets could be tested if they are also affected.

To address this interesting question, we investigated whether other miR-134 targets are regulated by Ago2 phosphorylation at S387. We co-expressed our Ago2 molecular replacement constructs with luciferase reporters incorporating the 3'UTRs of *PUM2* and *CREB1*, which have been shown previously to contain miR-134 binding sites (Fiore et al., 2014). Interestingly, expression of these reporters was unaffected by Ago2 S387 phospho-mutants, indicating that not all miR-134 targets are regulated by Ago2 phosphorylation at S387. These new data are presented in Fig. 5 of the revised manuscript. These results suggest the existence of specific features in the *LIMK1* 3'UTR, which are distinct from the *PUM2* and *CREB1* 3'UTRs, allowing *LIMK1* to be regulated by Ago2 phosphorylation at S387. We feel that identifying these specific sequences within the *LIMK1* 3'UTR would require a large amount of work that is beyond the scope of the current manuscript. We have included a discussion of this point on P.18 of the revised manuscript.

References:

- Calabrese, B., Saffin, J.M., and Halpain, S. (2014). Activity-dependent dendritic spine shrinkage and growth involve downregulation of cofilin via distinct mechanisms. *PLoS One* 9, e94787.
- Fiore, R., Rajman, M., Schwale, C., Bicker, S., Antoniou, A., Bruehl, C., Draguhn, A., and Schratt, G. (2014). MiR-134-dependent regulation of Pumilio-2 is necessary for homeostatic synaptic depression. *EMBO J.* 33:2231-46.
- Meng, Y., Zhang, Y., Tregoubov, V., Janus, C., Cruz, L., Jackson, M., Lu, W.Y., MacDonald, J.F., Wang, J.Y., Falls, D.L., et al. (2002). Abnormal spine morphology and enhanced LTP in LIMK-1 knockout mice. *Neuron* 35, 121-133.
- Nakamura Y., Wood C.L., Patton A.P., Jaafari N., Henley J.M., Mellor J.R., and Hanley J.G. (2011). PICK1 inhibition of the Arp2/3 complex controls dendritic spine size and synaptic plasticity. *EMBO J.* 30, 719-30.
- Rajgor, D., Fiuza, M., Parkinson, G.T., and Hanley, J.G. (2017). The PICK1 Ca²⁺ sensor modulates N-methyl-d-aspartate (NMDA) receptor-dependent microRNA-mediated translational repression in neurons. *J Biol Chem* 292, 9774-9786.
- Storchel, P.H., Thummler, J., Siegel, G., Aksoy-Aksel, A., Zampa, F., Sumer, S., and Schratt, G. (2015). A large-scale functional screen identifies Noval1 and Ncoa3 as regulators of neuronal miRNA function. *EMBO J* 34, 2237-2254.
- tom Dieck, S., Kochen, L., Hanus, C., Heumüller, M., Bartnik, I., Nassim-Assir, B., Merk, K., Mosler, T., Garg, S., Bunse, S., Tirrell, D.A., and Schuman, E.M. (2015). Direct visualization of newly synthesized target proteins in situ. *Nat Methods* 12, 411-4.

Thank you for submitting your revised manuscript to The EMBO Journal. Your study has now been re-reviewed by the three referees and their comments are provided below. As you can see below, the referees appreciate the changes and support publication here. Referee #2 finds that there is not enough data to support that the physiological relevance of Ago2-S387 - see comments below. I think this issue can be addressed with a careful discussion. Let me know if we need to discuss this further.

REFeree REPORTS

Referee #1:

The authors have satisfactorily addressed all my previous concerns. I can therefore recommend publication of this manuscript.

Referee #2:

In the revised manuscript, the authors have performed new experiments and overall addressed my concerns.

Although the new experiments are informative, the physiological relevance of Ago2-S387 is unclear. The function of Ago2-S387 was investigated in CA3-CA1-LTD where the authors found that phosphorylation of Ago2-S387 is not required for this form of plasticity. The result showed that both S387A and S387D-Ago2 were functional and rescued the phenotype caused by Ago2-KD. The authors interpreted this result as different mechanisms regulating structural and functional plasticity, but there is no evidence shown for structural plasticity at CA3-CA1 synapses. Therefore it is unclear whether phosphorylation of S387 plays a physiological role in NMDAR-dependent LTD. Since this is the major claim of the paper, I suggest the authors either remove this claim from the manuscript or find another NMDAR-dependent plasticity where Ago2-S387A can not be functional.

My other points have been thoroughly addressed.

Referee #3:

In their revised manuscript, the authors have addressed all points that I had raised on their previous version. They have performed additional experiments that solidify their conclusions. I am satisfied with the revised manuscript and do not have any extra points or comments.

Response to referee 2, manuscript EMBOJ-2017-97943R1. Rajgor et al.

We thank the referee for the constructive comment, which we have addressed below.

Although the new experiments are informative, the physiological relevance of Ago2-S387 is unclear. The function of Ago2-S387 was investigated in CA3-CA1-LTD where the authors found that phosphorylation of Ago2-S387 is not required for this form of plasticity. The result showed that both S387A and S387D-Ago2 were functional and rescued the phenotype caused by Ago2-KD. The authors interpreted this result as different mechanisms regulating structural and functional plasticity, but there is no evidence shown for structural plasticity at CA3-CA1 synapses. Therefore it is unclear whether phosphorylation of S387 plays a physiological role in NMDAR-dependent LTD. Since this is the major claim of the paper, I suggest the authors either remove this claim from the manuscript or find another NMDAR-dependent plasticity where Ago2-S387A can not be functional. Our data demonstrate that phosphorylation of Ago2 at S387 is not directly involved in NMDAR dependent CA3-CA1 LTD. Nevertheless, we show that S387 phosphorylation is involved in the

regulation of basal synaptic transmission, demonstrating a physiological role for this signalling event. This is discussed in full on P.19 of the manuscript.

The major claim of the paper is that phosphorylation of S387 plays a physiological role in NMDAR-dependent spine shrinkage, not LTD per se. However, we acknowledge that there were some instances of ambiguity on this issue in the manuscript, so we have removed such statements and/or re-worded the text to ensure clarity. These are in red text in the revised manuscript.

To define the role of S387 phosphorylation-dependent structural plasticity in a physiological context will require advanced 2-photon imaging, ideally in vivo. We believe that these experiments are beyond the scope of the current manuscript.

Corresponding Author Name: Jonathan Hanley

Manuscript Number: EMBOJ-2017-97943.